# Measurement report: Source attribution and estimation of black carbon levels in an urban hotspot of the central Po valley: An integrated approach combining high-resolution dispersion modelling and micro-aethalometers

Giorgio Veratti[1,2], Alessandro Bigi[2], Michele Stortini[3], Sergio Teggi[2], and Grazia Ghermandi[2]

[1]Department of Life Sciences. University of Modena and Reggio Emilia, 41125 Modena, Italy
[2]Department of Engineering 'Enzo Ferrari'. University of Modena and Reggio Emilia, 41125 Modena, Italy
[3]ARPAE, Regional Environmental Agency of Emilia-Romagna, 40122 Bologna, Italy
**Correspondence:** Giorgio Veratti (giorgio.veratti@unimore.it)

**Abstract.** Understanding black carbon (BC) levels and their sources in urban environments is of paramount importance due to their far-reaching health, climate and air quality implications. While several recent studies have assessed BC concentrations at specific fixed urban locations, there is a notable lack of knowledge in the existing literature on spatially resolved data alongside source estimation methods. This study aims to fill this gap by conducting a comprehensive investigation of BC levels and sources in Modena (Po valley, Italy), which serves as a representative example of a medium-sized urban area in Europe. Using a combination of multi-wavelength micro-aethalometer measurements and a hybrid Eulerian-Lagrangian modelling system, we studied two consecutive winter seasons (February-March 2020 and December 2020-January 2021). Leveraging the multi-wavelength absorption analyser (MWAA) model, we differentiate sources (fossil fuel combustion, FF, and biomass burning, BB) and components (BC vs. brown carbon, BrC) from micro-aethalometer measurements. The analysis reveals consistent, minimal diurnal variability in BrC absorption, in contrast to FF-related sources, which exhibit distinctive diurnal peaks during rush hours, while BB sources show less diurnal variation. The city itself contributes significantly to BC concentrations (52% ± 16%), with BB and FF playing a prominent role (35% ± 15% and 9% ± 4%, respectively). Long-distance transport also influences BC concentrations, especially in the case of BB and FF emissions, with 28% ± 1% and 15% ± 2%, respectively. When analysing the traffic related concentrations, Euro 4 diesel passenger cars considerably contribute to the exhaust emissions. These results provide valuable insights for policy makers and urban planners to manage BC levels in medium-sized urban areas, taking into account local and long-distance sources.

## 1 Introduction

Both black carbon (BC) and elemental carbon (EC) are carbonaceous particles associated with particulate matter (PM) that result from incomplete combustion processes and have light-absorbing properties. Although they are commonly mistaken for synonyms, these two terms should not be treated as completely interchangeable. BC denotes materials primarily composed of carbon with high light-absorption potential, while EC specifically denotes pure carbon without bonds to other elements

(Petzold et al., 2013). While these formal definitions strive for rigour, their practical application can be challenging due to the inherent variability of carbonaceous matter. Unlike pure substances, real-world BC exists as a complex mixture of various compounds with different properties. For this reason, a common approach to defining BC is based on properties such as light absorption, solubility, thermal stability, morphology or microstructure (Petzold et al., 2013).

BC originates from a variety of sources, including combustion engines (especially diesel), residential burning of wood and coal, field burning of agricultural residues, as well as forest and vegetation fires. The detrimental consequences associated with BC have become a key element of scientific investigation and public concern, necessitating a comprehensive understanding of its sources, behaviour, and potential mitigation strategies. The adverse effects of BC on human health are well-documented in the literature. Inhalation of BC particles has been linked to various respiratory and cardiovascular conditions, including asthma, chronic bronchitis, reduced lung function, and increased risk of heart attacks and strokes (Song et al., 2022; Janssen et al., 2012; Grahame et al., 2014; Rohr and McDonald, 2016; Janssen et al., 2011). Moreover, long-term exposure to BC has been associated with an elevated incidence of lung cancer (Chang et al., 2022). These health risks are particularly worrisome considering the widespread prevalence of BC in urban areas (Reche et al., 2011; Ali et al., 2021) and its ability to penetrate deep into the respiratory system (Saputra et al., 2014; Tang et al., 2020), posing a significant threat to vulnerable populations, such as children, the elderly, and individuals with pre-existing respiratory disorders.

BC not only poses adverse health effects but also exerts significant impacts on the Earth's radiation balance through various mechanisms (Chung and Seinfeld, 2005; Wang, 2004; Roberts and Jones, 2004). These include direct and indirect effects, altering the amount, distribution, and properties of solar energy (Team et al., 2023). The direct effects encompass the absorption and scattering of sunlight by BC, leading to localised atmospheric warming and influencing the distribution of solar radiation within the atmosphere (Menon et al., 2002; Ramanathan et al., 2001; Ramanathan and Carmichael, 2008). Estimates of BC's direct radiative forcing range from +0.1 to +1.0 $\mathrm{W\,m^{-2}}$, exhibiting variations depending on the methodology employed, such as bottom-up emission inventory and atmospheric transport model approaches, or observation-based estimates (Wang et al., 2016; Bond et al., 2013). In addition to the direct effects, BC particles have indirect impacts on the Earth's radiation balance by interacting with clouds. BC can serve as cloud condensation nuclei (CCN) or ice nuclei (IN), influencing the formation of cloud droplets and ice particles (Hendricks et al., 2011; Koch et al., 2011). The presence of BC can modify cloud microphysical properties, including droplet size, concentration, cloud lifetime and precipitation processes. These indirect effects have been estimated to range from -0.13 to -0.11 $\mathrm{W\,m^{-2}}$, contributing to a negative radiative forcing (Cherian et al., 2017; Koch et al., 2011). Considering both the direct and indirect effects, the combined impact of BC on the Earth's radiation balance results in a positive radiative forcing on the climate system, exacerbating global warming and climate change.

In the literature, several approaches have been employed to measure BC, EC or more generally light-absorbing aerosols. These approaches can be categorised into four main methods: thermo-optical determination, photothermal interferometry, photoacoustic spectroscopy and optical determination. Thermo-optical determination involves the measurement of light-absorbing aerosols by analysing their thermal properties. The technique typically relies on heating the sample in a controlled environment and measuring the resulting changes in optical properties, such as light absorption and scattering, as a function of temperature. By observing these changes, one can estimate the amount and type of light-absorbing aerosols present in the sample (Chow

et al., 2007; Bauer et al., 2009; Brown et al., 2019). Field experiments adopting this determination method are for example Merico et al. (2019); Liakakou et al. (2020); Bigi et al. (2017).

Photo-thermal interferometry is a technique that combines laser-induced heating with interferometric measurements to quantify light-absorbing aerosols (Lee and Moosmüller, 2020; Visser et al., 2023; Li et al., 2016). By directing a laser beam onto the aerosol sample, the absorbed light generates local temperature gradients, causing changes in the refractive index and thus altering the phase of a probe laser beam passing through the sample. These phase changes are then measured interferometrically to determine the concentration of light-absorbing aerosols. On the other hand, photo-acoustic methodologies involve the use of laser-induced acoustic waves to detect and quantify light-absorbing aerosols. In this technique, a pulsed laser is used to heat the sample, causing the aerosols to absorb the light and generate acoustic waves. By measuring the amplitude of these acoustic waves, the concentration of light-absorbing aerosols can be determined (Fischer and Smith, 2018; Petzold and Niessner, 1996; Guo et al., 2014). The deployment of photo-thermal and photo-acoustic instruments in field settings has been limited due to operational challenges, which has hindered the widespread adoption of these techniques. However, there has been a recent resurgence of interest in these methods with the development of a new generation of photometers and interferometers (Drinovec et al., 2022; Visser et al., 2020, 2023).

In addition to the aforementioned techniques, optical determination has gained popularity in recent years for measuring the light absorption properties of BC aerosols due to its cost-effectiveness, comparability with chemical analysis and ability to provide high temporal resolution results for real-time monitoring studies (Kaskaoutis et al., 2021a, b; Yus-Díez et al., 2021; Bernardoni et al., 2021). In this approach, sample air is passed through a filter tape where aerosol particles are collected. Optical filter photometers measure light transmission, or a combination of reflection and transmission, through the sample-loaded filter and calculate the attenuation coefficient from the rate of change of attenuation over time. However, the attenuation coefficient can differ significantly from the true aerosol absorption coefficient due to two main artefacts. The first is the enhancement of the optical path, and hence the enhancement of light absorption of the deposited particles, due to the multiple scattering of the light beam at the filter fibres and between particles and fibres. The second is the loss of instrument sensitivity with increasing particle loading. To overcome these limitations, several empirical corrections have been proposed in the literature. For instance, the $C_{ref}$ factor is used to correct for the multiple scattering effect, while the $f(ATN)$ function is applied to compensate the loading effect. Typically, the $C_{ref}$ factor is assumed a priori, but it can also be determined experimentally through multi-instrument colocation. Conversely, the loading correction function $f(ATN)$ is increasingly estimated online using dual-spot technology (Drinovec et al., 2015), or it can be estimated offline using dedicated algorithms (Weingartner et al., 2003; Virkkula et al., 2007; Park et al., 2010). The specific mass absorption efficiency (MAE; also known as the mass absorption cross section, MAC), can then be used to convert the aerosol light absorption coefficient into the light-absorbing carbon mass concentration. Although this relationship is simple, the value of the MAE varies considerably in time and space depending on emission sources, transport phenomena, combustion conditions, particle ageing and mixing state (Chan et al., 2011; Mbengue et al., 2021), making the conversion process a source of potential uncertainty (Petzold et al., 2013). When considering BC material, the term equivalent BC (eBC) is commonly used to report the mass concentration indirectly determined by light absorption techniques such as filter-based absorption photometers (Petzold et al., 2013). Therefore, the term eBC will be used throughout

this study to refer to those calculated using filter-based absorption photometers.

BC exhibits distinctive light absorption characteristics, particularly in the infrared range, facilitating its identification and quantification. Widely used instruments exploiting this technique are the aethalometer, such as the AEs models (e.g. AE31, AE33, AE36 and AE43, Magee Scientific Co.) or the portable micro-aethalometer MA series (e.g. MA200, MA300 and MA350, Aethlabs) which are filter-based optical instruments operating at different wavelengths. This later feature, coupled with the usage of source-specific absorption Ångström exponent, can be used for the source apportionment of black carbon (Zotter et al., 2017; Sandradewi et al., 2008; Massabò et al., 2015; Bernardoni et al., 2017b).

Source apportionment of BC is essential to gain insight into the specific pollution sources contributing to atmospheric concentrations. This analysis enables the development of targeted mitigation strategies, in line with the recommendation proposed by the EU legislation (European Council, 2008), which highlights the importance of identifying and addressing specific emission source sectors to effectively reduce air pollution. Furthermore, the World Health Organization (WHO), in its recently updated global air quality guidelines, has recommended the systematic measurement of BC (or EC), the establishment of BC inventories and, where appropriate, the implementation of BC reduction measures (WHO, 2021). In 2022, the European Commission proposed revisions to the Ambient Air Quality Directives to bring European Union air quality standards more closely in line with WHO recommendations (European Council, 2022). These revisions highlight the importance of monitoring emerging pollutants such as BC to support scientific understanding of their effects on health and the environment, in line with the WHO's guidance. This research aims to provide information on the levels of BC and to identify the main sources of BC in the urban environment of Modena, a city of about 200,000 inhabitants located in the middle of the Po valley, a densely populated and industrialised region in northern Italy, known for its high levels of air pollution (Bigi and Ghermandi, 2016; Bigi et al., 2012; Lonati et al., 2017; Perrino et al., 2014; Veratti et al., 2023; Thunis et al., 2021).

Previous studies have investigated the sources and transport of particulate matter in the Po valley using both chemical transport models (CTMs) and aerosol composition, such as for example Scotto et al. (2021); Paglione et al. (2020); Bernardoni et al. (2011); Pepe et al. (2019); Belis et al. (2019); Bernardoni et al. (2017a), but there is still a knowledge gap regarding the contribution of different sources to BC concentrations. For instance, a study by Mousavi et al. (2019) combined the usage of aethalometer, thermo-optical instruments and [14]C analysis to apportion between fossil fuel and biomass burning for three sites in the urban area of Milan and its surrounding. Similar studies in the Po valley have been conducted employing a variety of instruments such as aethalometers, Multi-Angle Absorption Photometers (MAAP), polar-photometers, and independent measurement of levoglucosan (Bernardoni et al., 2021; Massabò et al., 2015; Bernardoni et al., 2017b; Gilardoni et al., 2020). In addition, concerns about health effects have led recently to a surge of research in urban areas, highlighting eBC concentrations as a fundamental tracer of pollution in cities (Segersson et al., 2017; Pani et al., 2020). As a result, several studies have investigated the variability of eBC mass concentrations and the associated source apportionment at multiple urban locations. Examples of this research in Europe are Savadkoohi et al. (2023); Liakakou et al. (2020); Grange et al. (2020); Helin et al. (2018); Minderytė et al. (2022); Titos et al. (2017); Singh et al. (2018). Although all these studies utilised accurate methods for eBC determination and source apportionment, their results were limited to specific monitoring sites, including rural or urban

background locations. As a consequence, they were unable to provide continuous spatial information across the entire city, which is an important information for consistent epidemiological analysis.

In addition to the aforementioned studies, other researchers have conducted modelling studies at the European level to assess the performance of air quality models in reproducing carbonaceous aerosol concentrations and their absorption properties, such as Mircea et al. (2019); Curci et al. (2019). Other works aimed at estimating BC concentrations at street level in small neighbourhoods of Paris, Brussels or Maribor using a state-of-art multi-scale modelling system, Street-in-Grid (Lugon et al., 2021) or a more simplified approach (Brasseur et al., 2015; Ježek et al., 2018). However, none of these studies focused on assessing total BC concentrations over an entire urban area at high resolution, or attempted to identify the contribution of different sources at monitoring sites.

In this study, we conducted a two-sampling period campaign during wintertime (February-March 2020 and between December 2020 and January 2021) in Modena, deploying two MA200 multi-wavelength micro-aethalometers at distinct urban locations, at traffic and background sites. The primary objectives were to determine total BC concentrations at both sites and to investigate the potential sources contributing to the observed concentrations. To support the identification of local and regional sources and to provide spatial information about BC concentrations, we employed a hybrid Lagrangian-Eulerian modelling system composed by GRAMM-GRAL (Oettl, 2015b, a, c) and the NINFA (Network dell'Italia del Nord per previsioni di smog Fotochimico e Aerosol, Vitali et al. (2023)) modelling suite, which is built upon the CHIMERE chemical transport model (Mailler et al., 2017; Menut et al., 2021). By combining observed and modelled data, we aimed to improve our understanding of the main emission sectors contributing to total BC levels for the development of effective mitigation strategies and to provide valuable insights for addressing environmental and health concerns related to BC.

The paper is structured as follows: section 2 provides an overview of the measurement campaign and the models that make up the hybrid system. Further details on the emission estimation are given in subsection 2.5. Section 3 presents the statistical metrics used to validate the model performance against the observational data. Section 4 presents the results of the measurement campaign and modelling activities. Finally, section 5 presents some concluding remarks.

## 2 Materials and methods

### 2.1 Sampling sites and measurement description

The measurement campaign was carried out in Modena, a medium-sized city in the Emilia-Romagna region of northern Italy. Modena is located about 50 kilometres south of the Po River, in the Po valley, an area known for its dense population and industrialisation. The city, like the whole valley, has a continental climate with marked seasonal variations. In particular, during the winter months, the Po valley has specific meteorological characteristics, such as stable atmospheric conditions and frequent low wind speeds, which contribute to the occurrence of thermal inversions. These atmospheric phenomena can hinder the dispersion of air pollutants in the region, as reported by many studies, such as Bigi et al. (2012); Pernigotti et al. (2012a, b); Ghermandi et al. (2017). In addition, the Po valley is geographically bounded by the Apennines to the south and the Alps to the north, which act as natural barriers that further limit the movement of pollutants. As a result, the combination of high traffic

volumes, industrial activities and the large number of densely populated urban areas in the valley can lead to the accumulation of pollutants.

In this study, aerosol measurements were performed using two MA200 micro-aethalometers (microAeth® MA Series, Aethlabs, USA) equipped with PTFE filter tapes. These instruments quantify the absorption of light-absorbing carbonaceous aerosols at five different wavelengths (375, 470, 528, 625 and 880 nm), allowing for the estimation of eBC concentrations by employing wavelength-specific mass absorption efficiencies (MAE).

Sampling was performed at two air quality monitoring stations in the city, utilising the existing glassware manifold inlet lines used for reactive gas monitors. The inlet temperature was maintained at $30 \pm 2$ °C and no size cut-off was applied to the aerosol particles entering the devices. One station is located in a public park on the west side of the city, representing typical background conditions where pollution levels are influenced by a mixture of contributions rather than a single source type, while the other station is located in an urban traffic area characterised by high vehicular activity, serving as a representative location for areas of dense traffic and associated air pollution. The locations of the two stations are shown in Figure 1.

Throughout the study, the instruments operated at a flow rate of $100 \ \mathrm{ml min^{-1}}$ in dual-spot sampling mode to compensate for filter loading effects (Drinovec et al., 2015). Although several papers in the literature have emphasised the need for a multi-instrument approach to determine the multi-scattering correction factor ($C_{ref}$) for aethalometer filters (Bernardoni et al., 2021; Yus-Díez et al., 2021; Ferrero et al., 2021), with photothermal interferometry being recently identified as a suitable technique to provide a reference measurement of this parameter (Drinovec et al., 2022), this study lacked multi-instrument co-location. As a result, we opted for a constant $C_{ref}$ of 1.3 as suggested by the manufacturer to mimic the response of the AE33 aethalometer (Aethlabs, 2024). In addition, to convert the aerosol light absorption coefficient to equivalent mass concentration, we relied on the wavelength-specific MAE values provided in the MA200 reference manual and reported in Table S1. In support of these decisions, a recent instrument intercomparison performed at an urban background site in Athens (Stavroulas et al., 2022) showed limited differences in terms of eBC concentrations between a MAAP and the two MA200 units used in this study, set with $C_{ref}$ of 1.3 and default MAE values (linear slope of 1.00 in winter and 1.07 in summer, with an $r^2$ of 0.92 for both the seasons). From the same intercomparison campaign, two MA200 units were also compared with an AE33 aethalometer, showing strong agreement during winter (linear slope between 0.91 and 0.97 and $r^2$ of 0.97 for both devices). Furthermore, recent studies corroborate the results of the Athens intercomparison showing consistent findings when comparing the AE33 and the MA200. For example, Blanco-Donado et al. (2022) observed a linear regression slope of 0.97 and a $r^2$ of 0.93 during a 3-day intercomparison campaign in a suburban area of Barranquilla (Colombia). Similarly, Khan et al. (2024) reported a linear regression slope of 0.986 and a $r^2$ of 0.97 for a 14-hour intercomparison conducted in an urban background of Vilnius (Lithuania).

Table S1 in the supplementary material presents a comparison of the MAE values used in this study with those measured across various European locations. Although the values used for Modena are generally higher than those reported in the literature, they are consistent with previous observations conducted in the Po valley, such as Gilardoni et al. (2020) and Mousavi et al. (2019). Furthermore, the MAE value at 880 nm used for this study is in the range of values reported for other European urban

areas, such as Athens (Savadkoohi et al., 2024), Zurich and Bern (Grange et al., 2020), or in the range observed for urban traffic and background sites in Leipzig and Prague for MAE values at 637 nm (Savadkoohi et al., 2024).

Measurements at the urban traffic site were made during two periods: from 4 February to 7 March 2020 and from 26 December 2020 to 21 January 2021, with a time resolution of 1 minute, while the background site was monitored from 4 February to 7 March 2020 with a time resolution of 1 minute and from 26 December 2020 to 7 January 2021 with a time resolution of 5 minutes. To account for occasional low absorption readings at the background site, the 1-minute data at the urban background location were aggregated to 5 minutes using the dual-spot compensation algorithm implemented in the R

language (Bigi et al., 2023) following the formulation proposed by Drinovec et al. (2015). The MA200 measurements were evaluated based on the instrument's reported status, and flow calibration was performed before each filter change.

Strict lockdown measures were implemented in Northern Italy due to the SARS-CoV-2 pandemic from 8 March 2020 to 4 May 2020. Therefore, the winter data presented in this study represent a business-as-usual scenario, spanning across two winter seasons. Prior to the analysis, the concentration data were averaged to 1 hour to match the time resolution of observed

meteorological variables and align with the typical time step used in modelling applications at urban and regional scales.

Absorption measurements from the MA200 devices were further partitioned into specific components, including BC, BrC (hereafter noted as $b_{abs}^{BC}$ and $b_{abs}^{BrC}$) and their respective sources, namely fossil fuel (FF) and biomass burning combustion (BB), referred to $b_{abs,BC}^{FF}$ and $b_{abs,BC}^{BB}$, respectively. This partitioning was achieved using the Multi-Wavelength Absorption Analyzer model (MWAA) developed by Massabò et al. (2015); Bernardoni et al. (2017b). The MWAA model assumes that the

210 Absorption Ångström Exponent ($\alpha$), as defined by Moosmüller et al. (2009), is equivalent for BC and fossil fuel sources, while biomass burning is considered as the only source of BrC. Under these assumptions, the model holds that the total absorption at each wavelength adheres to the following equations:

$$b_{abs} = b_{abs}^{BC}(\lambda) + b_{abs}^{BrC}(\lambda) = A \cdot \lambda^{-\alpha^{BC}} + B \cdot \lambda^{-\alpha^{BrC}} \qquad (1)$$

$$b_{abs} = b_{abs}^{FF}(\lambda) + b_{abs}^{BB}(\lambda) = A' \cdot \lambda^{-\alpha^{FF}} + B' \cdot \lambda^{-\alpha^{BB}} \qquad (2)$$

In Equations 1 and 2 are defined with $\alpha^{BC} = \alpha^{FF} = 1$, which approximates the centre of the $\alpha$ probability density function (PDF) during the morning rush hour at the urban traffic site (Figure S1 in the Supplement), indicative of fresh uncoated BC particles, as shown by Liu et al. (2018). The $\alpha$ value for BrC was determined by a preliminary nonlinear fit of Eq. 2, with $\alpha^{BrC}$ treated as a free parameter (resulting in a mean $\alpha^{BrC} = 3.9$). $\alpha^{BB} = 2$ was set based on the upper tail of the $\alpha$ PDF calculated

from the fit over the five wavelength and applying a stringent filter of $r^2 > 0.99$ (Figure S1), as suggested for example by Tobler et al. (2021). The same value also aligns with the existing literature data for the Po valley (Costabile et al., 2017b; Vecchi et al., 2018; Bernardoni et al., 2011). Parameters A and B were then derived for each sample by multi-wavelength fitting of Eq. 1 (with a fixed $\alpha^{BrC}$ value) and A' , B' by multi-wavelength fitting of Eq. 2. For a comprehensive understanding of the MWAA model, reference can be made to the works of Massabò et al. (2015) and Bernardoni et al. (2017b). In addition, for a more

detailed information on the source and component assignment performed on the data used in this study, detailed information

**Table 1.** Characteristics and locations of meteorological and air quality stations employed in this study.

| Station name | label | Longitude (°) | Latitude (°) | sensor height above the ground (m) | Type |
|---|---|---|---|---|---|
| Campus - DIEF | CMP | 10.95037 | 44.62832 | 10 | Meteorology |
| Dexter - Arpae | DEX | 10.91699 | 44.65639 | 40 | Meteorology |
| Geophysical observatory | OSS | 10.92981 | 44.64809 | 50 | Meteorology |
| Policlinico | POL | 10.94429 | 44.63580 | 40 | Meteorology |
| S. Pietro Capofiume | SPC | 11.62567 | 44.65467 | - (soundings) | Meteorology |
| via Giardini | urban traffic | 10.90572 | 44.63699 | 4 | Air Quality |
| parco Ferrari | urban background | 10.90731 | 44.65157 | 4 | Air Quality |

can be found in the companion study by Bigi et al. (2023). Meteorological variables essential for the reconstruction of the wind field by the GRAMM-GRAL model, including downward global radiation, wind speed and direction, were obtained from the meteorological network of the Osservatorio Geofisico di Modena (identified as CMP, OSS and POL stations) and from the monitoring network managed by the Regional Environmental Agency Arpae (represented by the DEX station). Table 1 shows the locations and characteristics of the stations used in this study, while Figure 1 shows their positions on a map. The table also includes information on the locations where soundings were carried out and subsequently used in this study, denoted by the SPC label.

## 2.2 The hybrid Eulerian-Lagrangian modelling system

In this study, one of the main objectives was to perform a spatial investigation of BC concentrations in the urban environment of Modena and to provide valuable information on source apportionment. To achieve this, we used a hybrid modelling approach based on two key components: the GRAMM-GRAL dispersion modelling suite, a state-of-the-art Lagrangian simulation system, and the NINFA modelling tool.

The GRAMM-GRAL suite was specifically selected for its ability to simulate the dispersion of emissions within the study area, taking into account the complex urban landscape with its various obstacles such as buildings and structures. On the other hand, NINFA played a crucial role in complementing the output of the Lagrangian model and providing additional insights into BC pollution. NINFA was used to estimate background concentrations from sources outside the urban area and to assess the influence of regional factors on BC levels in Modena. By integrating the strengths of both models, we aimed at a comprehensive understanding of BC concentrations in the study area, capturing the complex dynamics of BC transport in the urban environment. Furthermore, the Lagrangian models used in our approach offered the advantage of generating concentration fields specific to selected source emissions, allowing the estimation of source apportionment and providing valuable information on the contributions of different emission sectors to BC levels. To avoid duplication of emissions in Modena, the BC emission fluxes within the urban domain of interest were set to zero when using NINFA. In addition, BC was treated as an inert species in both models, leading to the exclusion of any chemical reactions involving BC. As a result, the total BC concentration was determined by combining the NINFA output, which represents the contribution of the external areas of Modena, with the GRAL output, which instead represents the contribution of the city. For a more complete understanding of the modelling system and the coupling between an Eulerian and a Lagrangian model used in this study, reference is made to previously published works such as Veratti et al. (2020, 2021), where a detailed description is provided.

## 2.3 NINFA set-up

Simulations were conducted using NINFA, a modelling system based on the state-of-the-art chemical transport model CHIMERE v2017r4 (Mailler et al., 2017) and operated by the Regional Agency for Prevention, Environment and Energy (ARPAE). This suite was employed to investigate surface concentrations of BC aerosol at the regional scale. The study area covered a significant part of the Po valley, with a grid size of 114 x 86 cells. The grid extended from 8.78° E to 13.13° E longitude and from 43.57° N to 45.79° N latitude, with a fine horizontal resolution of 3 km (see Figure 1 for visualisation). The vertical grid consisted of 15 levels, ranging from 997 hPa (25 m) for the first layer to 500 hPa. This high-resolution grid was specifically chosen to accurately represent the spatial scale characteristic of regional air pollution in the area of interest, and it also allowed a precise delineation of the urban area of Modena, where urban emissions were excluded from the simulations. The meteorological input to NINFA was provided by the output of COSMO-2I, a limited domain atmospheric model used by the Italian National Civil Protection Agency and operated by the local environmental agency Arpae. The COSMO domain covers the Italian peninsula with a horizontal resolution of 2.2 km and 30 vertical levels ranging from 20 m to 22 km. Meteorological variables were spatially interpolated to the NINFA grid using the libsim libraries (Clima, 2023). The aerosol module imple-

mented within CHIMERE provided hourly concentrations of 6 chemical species including sulphates, nitrates, ammonium, primary and secondary organic particles, dust and BC. Aerosols were represented using a size bin approach with 10 bins with mean mass median diameters ranging from 10 nm to 40 μm, and BC was simulated with a mass distribution centred at 200 nm with a sigma of 1.2, consistent with previously reported experimental studies (Schwarz et al., 2008; Ning et al., 2013; Li et al., 2023). Detailed information on the gas phase chemistry and secondary organic aerosol formation scheme used in this study (which is not directly relevant to the ultimate purpose of the current research) can be found in Veratti et al. (2023). In our investigation, we focused only on BC, which was treated as an inert species. Other parameterisations include the van Leer scheme (Van Leer, 1977) scheme for horizontal transport and the Troen and Mart formulation (Troen and Mahrt, 1986) for vertical turbulent mixing. In addition, wet and dry deposition processes of aerosols were considered using the equations of Henzing et al. (2006) and Zhang et al. (2001), respectively.

Boundary conditions were obtained from the national model kAIROS (AIR Operational System), which is the air quality forecast model operated by the regional air quality agency Arpae on a daily basis throughout Italy (Stortini et al., 2020), using the same version of CHIMERE used in this study with a spatial horizontal resolution of 7 km. The simulation period chosen for our study was from 4 February to 7 March 2020 and from 26 December 2020 to 21 January 2021, corresponding to the availability of BC measurements in Modena. In order to ensure the reliability and accuracy of our simulations, we implemented a spin-up period of 3 days in order to reach an equilibrium state and to establish stable initial conditions before the periods of interest.

In order to perform a source-tagged BC simulation aimed at distinguishing the contribution to total BC concentrations from different sources, we made modifications to the original CHIMERE code. These modifications allowed the specific allocation of the contribution of different sectors coming from outside the urban area of Modena.

The performance of CHIMERE in reproducing gaseous species, total PM and PM components has been extensively evaluated in several intercomparison studies. Notably, the EURODELTAIII (Bessagnet et al., 2016; Mircea et al., 2019), AQMEII (Pirovano et al., 2012), and POMI (Pernigotti et al., 2013) exercises have been significant in this context. Specifically, Mircea et al. (2019) focused on the simulation of carbonaceous aerosols over Europe. When EC concentrations within the $PM_{2.5}$ matrix were compared with corresponding measurements, CHIMERE showed satisfactory performance in reproducing the observed trend for the years 2006-2009, with a Pearson's correlation coefficient (r) between modelled and measured concentrations ranging from 0.6 to 0.85, a normalised centred Root Mean Square Error (RMSE) between 1.5 and 2.0 and a normalised Standard Deviation (SD) between 0.05 and 1.25.

In the EURODELTAIII exercise (2006-2009), CHIMERE generally overestimated $O_3$ concentrations compared to other participating models, with a Mean Bias (MB) between 6.3 and 22.5 $\mu g\,m^{-3}$. However, in 2009, all models, including CHIMERE, underestimated the measured concentrations due to biased boundary conditions. Despite this, CHIMERE had the lowest correlation coefficient (ranging from 0.27 to 0.71) but also the lowest RMSE among the models. For $NO_2$, its performance was comparable to other models, with r ranging from 0.67 to 0.72 and MB ranging from -0.64 to 0.64 $\mu g\,m^{-3}$, depending on the simulation year. For $SO_2$, the correlation coefficient ranged from 0.2 to 0.4, similar to other models, but CHIMERE was closest to the observations (MB from -0.46 to 0.13 $\mu g\,m^{-3}$). For $PM_{10}$, CHIMERE generally underestimated the measurements, as did other models, with MB ranging from -6.59 to -0.52 $\mu g\,m^{-3}$, but it achieved the highest correlation coefficient (0.7). Similarly,

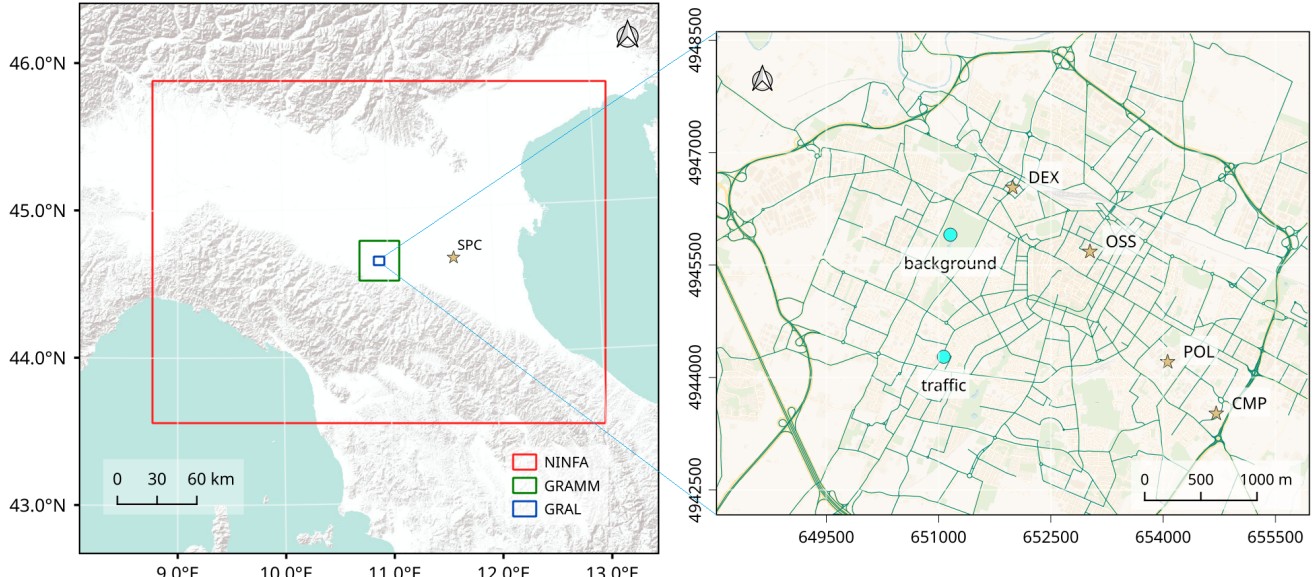

**Figure 1.** Spatial representation of the NINFA and GRAMM domains on the left (from Esri, USGS, NOAA), and the GRAL investigation domain on the right. The GRAL domain includes the positions of two urban air quality stations denoted by lightblue dots, specifically one located at the urban traffic (traffic) site and another at the urban background (background) site. Additionally, four meteorological stations used for meteorological validation and for selecting the meso-scale meteorological situation are indicated by yellow stars within the GRAL domain.

CHIMERE performed well for $PM_{2.5}$ over all simulated years, with the lowest RMSE (6.59 µg m$^{-3}$) and a comparable MB (between -1.00 and -2.39 µg m$^{-3}$) to the other models.

In addition, in the POMI intercomparison exercise, CHIMERE was used alongside EMEP, AURORA, CAMx, TCAM, and REM-CALGRID to simulate $O_3$ and PM over the Po Valley. The results indicated that CHIMERE's performance was compa-

rable to the other models. For daily $PM_{10}$, CHIMERE had one of the lowest RMSE values (29.1 µg m$^{-3}$) and the highest correlation coefficient (0.6), with other statistical metrics in good agreement with the other models. However, for $O_3$, CHIMERE had the highest absolute MB (-9.1 µg m$^{-3}$) but also one of the lowest RMSE values (31.5 µg m$^{-3}$), with other metrics in agreement with the other models.

## 2.4   GRAMM-GRAL description and set-up

The GRAMM-GRAL modelling system is a comprehensive approach designed for simulating air pollution in urban areas. It integrates two tools: GRAMM (Graz Mesoscale Model) and GRAL (Graz Lagrangian Model). While GRAMM focuses on the simulation of meteorological variables such as temperature, humidity, wind speed, and wind direction, GRAL is a Lagrangian particle dispersion model capable of reproducing pollutant concentrations in the atmosphere. This system is particularly well suited for simulating air pollution in urban environments, even in the presence of steep topography.

GRAMM is a non-hydrostatic model that addresses the conservation equations for mass, enthalpy, momentum, and humidity. It takes into account the effects of topography and surface interactions, such as the exchange of heat, momentum, humidity, and radiation, across different land use categories. To handle turbulence, an algebraic turbulence model (Pandolfo, 1969) is used when Gradient Richardson Numbers exceed 0.3, while a k- approach is used for lower values. Standard vertical profiles of wind and temperature are used to initiate and drive GRAMM at its boundaries, depending on the synoptic scale forcing and stability class chosen as input. The initialization process follows the Pasquill-Gifford classification and a detailed theoretical formulation and a practical implementation can be found in the works of Almbauer et al. (1995); Oettl (2015b). The ability of GRAMM in reproducing meteorological conditions with different topographic settings and resolutions has been demonstrated in previous studies such as, Almbauer et al. (1995); Oettl (2015b); Thunis et al. (2003); Oettl (2021); Oettl and Reifeltshammer (2023). Furthermore, its performance has recently been evaluated by comparison with the widely used WRF model (Skamarock et al., 2008), showing very similar results for wind and temperature fields compared to observations (Oettl and Veratti, 2021).

Within GRAMM, the micro-scale model GRAL is embedded through a one-way coupling from the larger to the smaller scale. GRAL offers the option to operate in either diagnostic or prognostic mode. In the diagnostic mode, the flow field around buildings is determined by interpolating the large scale wind fields from GRAMM output assuming a logarithmic wind profile close to urban surfaces, then a Poisson equation is applied to correct velocities and guarantee the mass conservation. On the other hand, the prognostic mode, despite its higher computational cost, ensures superior accuracy by explicitly computing the flow through forward integration of the Reynolds-Averaged Navier-Stokes equations (RANS; Oettl (2015c)), while turbulence is estimated using a standard $k - \epsilon$ approach, neglecting Coriolis and buoyancy forces (Oettl, 2015c).

In this research, a cascade of scales was used to determine wind patterns within the city, from the synoptic to the building level. The study started with the simulation of large-scale meteorological conditions in the region surrounding the city of interest. An area of 30 km x 30 km was considered, centred on Modena, with a resolution of 200 m (see Figure 1). The simulation was carried out using GRAMM version 22.09, taking into account the local topography obtained from the Geoportale-Emilia-Romagna (2023) and land use data extracted from the Corine land cover database, updated to 2018 (CCL, 2018). These mesoscale flow patterns were used as the driving force for the computation of high-resolution winds within the city, specifically accounting for buildings and street canyons. Finally, Lagrangian dispersion calculations were performed using the dispersion module of GRAL (version 22.09), with constraints imposed by the previously generated micro-scale wind fields. The GRAL domain includes the historical city centre, the ring road and most of the urban environment of Modena, covering a domain of approximately 7.9 km x 6.5 km, with a horizontal resolution of 4 m (see Figure 1).

Since the computation of the city-scale wind field at 4 m resolution over an entire urban environment is computationally intensive, the approach of Berchet et al. (2017) was implemented using a discrete representation of different weather situations. This approach took into account the classification of atmospheric stability, large-scale wind speed and direction at the boundaries of the domain into discrete categories. Large-scale wind directions were divided into 36 classes of 10 degrees each, while wind speeds were classified into 15 classes ranging from 0.25 to 7 $\mathrm{m\,s^{-1}}$. Atmospheric stability was classified according to the the Pasquill-Gifford classes (EPA, 2000). This resulted in a catalogue of about 1100 physically meaningful reference weather situations that occurred during the two periods of interest. Once the catalogue of meteorological situations

was completed, at each simulation time step the wind field was selected by matching the meteorological catalogue with the observed meteorological conditions at the reference urban stations within the urban area of Modena (see Figure 1). Lagrangian dispersion calculations were then performed at hourly intervals in prognostic mode, using the particle diameter of 2.5 μm. In order to validate the model wind fields at the urban scale, we used the same stations that were used to select the wind field situation from the catalogue. The location of the stations is shown in Figure 1, while their characteristics are reported in Table 1. On the other hand, the validation of the results obtained with NINFA and GRAL was carried out using eBC measurements at two sites, respectively urban background and traffic sites.

## 2.5 Anthropogenic emissions

### 2.5.1 City scale

The anthropogenic emissions used in this study were obtained from the regional emission inventory for the Emilia-Romagna region, regularly compiled by the local environmental agency Arpae (INEMAR, 2023). This inventory provides annual totals of $PM_{10}$, $PM_{2.5}$, $NO_x$, CO, $SO_2$, $NH_3$ and non-methane volatile organic compounds (NMVOCs) for each municipality, in this case specifically for the year 2017. For transport emissions, however, we used a bottom-up approach rather than relying solely on the regional inventory. The latter has been shown to be successful in reporting high-resolution traffic emissions in previous studies conducted in the same region (Ghermandi et al., 2019, 2020; Veratti et al., 2020, 2021).

To estimate traffic emissions at the city scale, we employed a comprehensive bottom-up approach that integrates emission factors (EF) and activity data. The activity data was derived from traffic flows simulated by the PTV VISUM model. Then, specific exhaust EF were applied to estimate the total emissions, taking into account various factors such as fleet composition, vehicle type, fuel, engine capacity, load displacement, road slope, Euro emission standard and average travelling speed. Additionally, we considered non-exhaust traffic EF, which are contingent upon vehicle weight and travelling speed, to estimate emissions from tire, brake and road wear.

To estimate BC exhaust emissions from road transport, we adopted the Tier 3 methodology outlined in the European EMEP/EEA guidelines (Ntziachristos and Samaras, 2019), which includes the calculation of both hot emissions, which occur when the engine is operating at its normal temperature, and emissions during transient thermal engine operation, commonly referred to as 'cold start' emissions. To determine the EF for each vehicle fleet category, we used flow velocity estimates from PTV VISUM, and the individual EFs were averaged taking into account the local fleet composition (ACI, 2023) and the total annual kilometres travelled by each vehicle category considered (ISPRA, 2023). As a result, we obtained two EFs per road segment: one for light vehicles (such as cars and mopeds) and another for light and heavy commercial vehicles. This estimate corresponds to the two classes available in the PTV VISUM output and allows a consistent representation of BC exhaust emissions.

On the other hand, for the estimation of non-exhaust BC traffic emissions, the Tier 2 approach described in Ntziachristos and Boulter (2019) was implemented. The calculation was based on the so-called "detailed methodology", which takes into account the speed dependency of tyre and brake wear emissions from moving vehicles, while for road surface wear the emissions depend

only on the number of vehicles travelling on a given road. The exhaust and non-exhaust calculation methods used in this study were coded in the R programming language and embedded in a package called VERT (Vehicular Emissions from Road Traffic;
Veratti et al., 2024). The VERT package provides a comprehensive framework for estimating emissions from road traffic, including exhaust, non-exhaust and evaporative emissions. The calculation process takes into account fleet composition and vehicle flows for cars, mopeds, light-duty vehicles and trucks or aggregated classes depending on the level of data available. This package includes not only the latest emission factors and methodologies recommended by the European EMEP/EEA guidelines for BC emissions, but also for a range of other pollutants including CO, NO, primary $NO_2$, $NO_x$, VOC, $CH_4$,
PM, Organic Carbon, $N_2O$, $NH_3$, $CO_2$ and $SO_2$. Furthermore, in recognition of the significant uncertainties associated with PM speciation in the determination of BC emission factors, VERT provides an uncertainty range for both exhaust and non-exhaust emission determination based on the speciation interval specified in the methodology documentation (Ntziachristos and Samaras, 2019; Ntziachristos and Boulter, 2019). In this study, we aimed to assess the impact of emission uncertainties on the model results and investigate the potential range of outcomes. To achieve this, we ran a series of scenarios incorporating
different hypotheses regarding BC speciation factors and the distribution of emission sources within the city boundaries. This approach allowed us to obtain a comprehensive set of BC concentration maps covering a plausible range of outcomes.

For traffic emissions, we ran a base case scenario using the reference speciation factor for BC as suggested by the European EMEP/EEA guidelines, and two additional scenarios using the lower and upper range values for BC speciation. In addition, to increase the comprehensiveness of our assessment of the overall impact of traffic, we extended the emission estimate beyond
exhaust and non-exhaust BC emissions and added resuspension due to traffic circulation. To include this source, we conducted five additional scenarios. Fuor of these were derived from the methodology proposed by Amato et al. (2012) and implemented as described in the HERMESv3 model (Guevara et al., 2020), while the fifth was evaluated based on the formulation of the NORTRIP model by Denby et al. (2013). Amato's approach relies on vehicle-specific emission factors, the values of which used in this study are detailed in Table 2. On the other hand, the NORTRIP formulation is based on ensuring mass conservation on
the road surface. For the latter reason, we used a three-step procedure to estimate the resuspension effects using the NORTRIP model. First, we calculated the dispersion of the full set of emissions, including traffic exhaust, non-exhaust and emissions from other activity sectors. Then we used the BC deposition on the road surface obtained from this calculation to estimate the resuspension effects based on the formulation reported below (Eq. 3). Finally, we performed another simulation using these emission rates to determine the contribution of resuspension to atmospheric concentrations. The calculation of the resuspension
rate $Q_{\mathrm{resusp}}$, which represents the amount of particles resuspended into the atmosphere according to the NORTRIP model, is determined according to Eq. 4, taking into account the mass deposited on the road surface $M_{\mathrm{dep}}$.

$$Q_{\mathrm{resusp}} = M_{\mathrm{dep}} \cdot F_{\mathrm{resusp}} \tag{3}$$

Where $F_{\mathrm{resusp}}$ is the resuspesion factor and can be calculated as:

$$F_{\mathrm{resusp}} = \sum_{v=1}^{2} N_{\mathrm{v}} \left( \frac{u_{\mathrm{v}}}{u_{\mathrm{ref(r)}}} \right) f_{0,v} \tag{4}$$

**Table 2.** Simulated emission scenarios for traffic sources.

| Label | Scale | Type | Light duty vehicles | Commercial vehicles |
|-------|-------|------|---------------------|---------------------|
| TRF-exh-1 | City | exhaust | average BC speciation factor[a] | average BC speciation factor[a] |
| TRF-exh-2 | City | exhaust | minimum BC speciation factor[a] | minimum BC speciation factor[a] |
| TRF-exh-3 | City | exhaust | maximum BC speciation factor[a] | maximum BC speciation factor[a] |
| TRF-nexh-1 | City | non-exhaust | average BC speciation factor[b] | average BC speciation factor[b] |
| TRF-nexh-2 | City | non-exhaust | minimum BC speciation factor[b] | minimum BC speciation factor[b] |
| TRF-nexh-3 | City | non-exhaust | maximum BC speciation factor[b] | maximum BC speciation factor[b] |
| TRF-res-1 | City | resuspension | $1.2 \text{ mg vehicle}^{-1}$ | $2.5 \text{ mg vehicle}^{-1}$ |
| TRF-res-2 | City | resuspension | $1.65 \text{ mg vehicle}^{-1}$ | $1.65 \text{ mg vehicle}^{-1}$ |
| TRF-res-3 | City | resuspension | $0.6 \text{ mg vehicle}^{-1}$ | $0.6 \text{ mg vehicle}^{-1}$ |
| TRF-res-4 | City | resuspension | $2.7 \text{ mg vehicle}^{-1}$ | $2.7 \text{ mg vehicle}^{-1}$ |
| TRF-res-5 | City | resuspension | NORTRIP model | NORTRIP model |
| TRF-bck-1 | Regional | exhaust, non-exhaust and resuspension | average BC speciation factor[a,b] | average BC speciation factor[a,b] |
| TRF-bck-2 | Regional | exhaust, non-exhaust and resuspension | minimum BC speciation factor[a,b] | minimum BC speciation factor[a,b] |
| TRF-bck-3 | Regional | exhaust, non-exhaust and resuspension | maximum BC speciation factor[a,b] | maximum BC speciation factor[a,b] |

[a] See table 3-91 of Ntziachristos and Samaras (2019) for the exhaust PM speciation factors for different vehicle technologies.

[b] See table 3-4 and table 3-6 of Ntziachristos and Boulter (2019) for non-exhaust emission factor range for different vehicle categories.

The variables in the Eq. 4 are defined as follows: $v$ represents the vehicle type, $N_v$ represents the vehicle flow (measured in vehicles per hour), $u_v$ represents the vehicle speed (measured in $\text{km h}^{-1}$), $u_{\text{ref(r)}}$ represents the reference vehicle speed for the resuspension process (measured in $\text{km h}^{-1}$), and $f_{0,v}$ represents the reference mass fraction of the resuspension process per vehicle. In line with previous studies by Thouron et al. (2018); Denby et al. (2013); Lugon et al. (2021), this study adopts the values $u_{\text{ref(r)}} = 50 \text{ km h}^{-1}$, $f_{0,\text{HDV}} = 5 \cdot 10^{-5}$ per vehicle, and $f_{0,\text{LDV}} = 5 \cdot 10^{-6}$ per vehicle. All traffic scenarios simulated in this study are summarised in Table 2.

A combination of speciation factors and detailed inventory data was used to estimate emissions from other activity sectors. As introduced at the beginning of this section, the Arpae inventory is used as a reference, while activity-dependent BC speciation factors reported by the EEA (2019) and listed in Table S2 in the Supplement were used to convert total PM emissions into BC estimates. In order to distribute the emissions within the urban area of Modena, land use data and building characteristics were used as proxy variables. However, it should be noted that in the Emilia-Romagna region there are still significant uncertainties regarding the distribution of emissions from biomass burning for non-industrial combustion. To address this limitation, we developed five different emission scenarios to capture the variability of emissions in terms of spatial distribution. These scenarios considered different source locations and different total emissions allocated to different areas of the city, including the historic centre, discontinuous urban fabric and rural areas. Table 3 summarises the characteristics of each scenario.

The daily temporal modulation of non-industrial combustion emissions is based on the concept of heating degree days, a metric developed to capture the energy demand required to heat a structure based on both external and internal temperatures. The formula used in this study to incorporate this calculation is derived from Mues et al. (2014). In addition to transport and

**Table 3.** Simulated emission scenarios for non-industrial combustion sources.

| Label | Scale | Emissions assigned to rural areas | Emissions assigned to discontinuous urban fabric | Emissions assigned to the historical city centre |
| --- | --- | --- | --- | --- |
| | | (%) | (%) | (%) |
| DMH-1 | City | 80 | 20 | 0 |
| DMH-2 | City | 60 | 40 | 0 |
| DMH-3 | City | 50 | 50 | 0 |
| DMH-4 | City | 50 | 40 | 10 |
| DMH-5 | City | 40 | 40 | 20 |
| DMH-bck-1 | Regional | 60 | 40 | included in discontinuous urban fabric |
| DMH-bck-2 | Regional | 40 | 60 | included in discontinuous urban fabric |
| DMH-bck-3 | Regional | 80 | 20 | included in discontinuous urban fabric |

biomass combustion, other relevant SNAP sectors included in the calculation are emissions from industry and other mobile machinery. However, due to their relatively small contribution in annual totals compared to transport and biomass burning, no additional scenarios were simulated to assess their impact range.

### 2.5.2 Regional scale

The emission data used for NINFA were taken from the same inventory used at the urban scale, where the total annual emissions for each pollutant are reported at the municipal level. To ensure a precise depiction at a more detailed spatial and temporal resolution, a downscaling procedure was implemented. Emissions were spatially allocated at the model resolution using the Corine land cover database (CCL, 2018) and temporally distributed using monthly, daily, and hourly profiles typical for northern Italy (Veratti et al., 2023). To avoid double counting of BC emissions within the urban area of Modena, the emission fluxes from this area were set to zero.

The availability of high resolution emission inventories produced by environmental agencies can sometimes be challenging. Therefore, a comparison between the annual BC emissions used in a study for the Emilia-Romagna region (INEMAR 2017) and common European and global emission datasets can provide valuable insights for future modelling studies in the Po valley. Table 4 presents a comparison between INEMAR 2017 (INEMAR, 2023; Marongiu et al., 2024), CAMS-REGv4.2 (2018; Kuenen et al., 2022), EDGARv6.1 (2018; Crippa et al., 2020) and EMEP (2017: Ullrich et al., 2023), categorised by sector. The comparison results show that for non-industrial combustion, EDGARv6.1 and EMEP agree quite well with INEMAR, with discrepancies between 8% and 10%, while CAMS-REGv4.2 is about 60% higher than INEMAR. For traffic emissions, the annual totals are very close to each other, with differences ranging between 3% and 22%, confirming that this sector is relatively well constrained. However, total emissions for other sectors exhibit significant discrepancies, partly due to the different classification systems used by different inventories to categorise emissions. Despite these differences, total BC emissions are relatively well aligned, with differences ranging from 11% to 17%.

**Table 4.** Comparison of annual BC emissions for the Emilia-Romagna region across different emission sectors as reported by INEMAR 2017, CAMS-REGv4.2, EDGARv6.1 and EMEP 2017.

| Area | Emission sector | INEMAR 2017 (tons) | CAMS-REGv4.2 (tons) | EDGARv6.1 (tons) | EMEP 2017 (tons) |
|---|---|---|---|---|---|
| | Non-industrial combustion | 672 | 1090 | 622 | 744 |
| | Industrial combustion | 30 | 65 | 154 | 68 |
| Emilia-Romagna region | Traffic | 532 | 549 | 586 | 680 |
| | Other mobile machinery | 298 | 109 | 1 | 300 |
| | Other sectors | 3 | 42 | 13 | 61 |
| | Total | 1535 | 1855 | 1376 | 1853 |

To assess the influence of external areas on BC concentrations in Modena, six scenarios were developed. Three scenarios focused on the speciation factors of both exhaust and non-exhaust traffic emissions. In particular, to convert the total PM emissions reported in the inventory into BC emissions, vehicle fleet dependent BC speciation factors calculated with the VERT package were used. The first scenario used average BC conversion factors, while the other two scenarios represented the lower and upper bounds of the BC speciation values. For non-industrial combustion BC emissions, three different spatial allocations were considered. In the first allocation, municipal totals were distributed with 40% allocated to rural areas and 60% to urban areas. In the second allocation, the same proportion was maintained, but 60% was allocated to rural areas and 40% to urban areas. In the third allocation, 80% was allocated to rural areas and 20% to urban areas. Table 3 provides an overview of each scenario simulated at the regional level.

## 3 Model evaluation

A number of statistical metrics have been used to evaluate the hybrid modelling system against observed data. These metrics include MB, Normalised Mean Bias (NMB), r, the proportion of predicted values within a factor of two of the observations called Factor of Two (FAC2), Normalised Mean Square Error (NMSE), Fractional Bias (FB), Normalised Average Difference (NAD) and RMSE. Detailed definitions of these metrics can be found in Section S3 of the Supplementary material.

## 4 Results

### 4.1 Measurements from MA200

Figure 2 shows the average diurnal variability of the aerosol absorption at 880 nm (panel a) from the two MA200 instruments partitioned using the results of the MWAA model, distinguishing between fossil fuels, biomass combustion and BrC. The same partitioning was also used to differentiate eBC concentrations (panel b of Figure 2) between fossil fuels (eBC-FF) and biomass combustion (eBC-BB), applying a fixed MAE (see Section 2.1). Due to the lack of in situ validation measurements to assess

the MAE of BrC and the significant uncertainty associated with its determination, see for example the diverse structures and properties of the wide range of chemical organic compounds, coupled with its sensitivity to variations in time and location, we chose not to convert $b_{abs}^{BrC}$ into concentrations. This decision was taken to avoid introducing additional uncertainty associated with assuming a specific MAE for BrC.

The measurements reported in Figure 2 were carried out during two different periods. The first period lasted from 4 February to 7 March 2020, while the second period spanned from 26 December 2020 to 21 January 2021 for the traffic site and from 26 December 2020 to 7 January 2021 for the urban background.

When analysing the results, it can be seen that $b_{abs}^{BrC}$ remain consistently very low on average throughout the day for both sites, averaging around $0.5\ \mathrm{Mm^{-1}}$ for the first period and around $1.2\ \mathrm{Mm^{-1}}$ for the second period. On the other hand, $b_{abs,BC}^{FF}$ shows the characteristic behaviour associated with traffic emissions, with two peaks during rush hours. Specifically, these peaks occur in the morning between 07:00 and 08:00 GMT and in the evening between 18:00 and 19:00 GMT or slightly delayed at the urban background site for the first period, between 18:00 and 20:00 GMT. This pattern is also observed for other traffic-related tracers such as NO, $NO_2$ and benzene measured at the same two sites, not shown here but reported in a companion study by Bigi et al. (2023).

In contrast, $b_{abs,BC}^{BB}$ shows less variability and generally lower diurnal absorptions compared to $b_{abs,BC}^{FF}$. In particular, their behaviour is different, with $b_{abs,BC}^{BB}$ reaching a minimum in the early afternoon, typically between 15:00 and 17:00 GMT. This phenomenon can be attributed to a combination of factors, including a higher mixing layer depth and lower emissions during these hours. In the later part of the afternoon, after 17:00 GMT, $b_{abs,BC}^{BB}$ typically increase and reach a maximum between 20:00 and 21:00 GMT, when the emitting sources are likely to be at their peak emission levels and the mixing layer depth is gradually decreasing due to surface cooling. Despite a likely decrease in the intensity of biomass burning emissions during the night, $b_{abs,BC}^{BB}$ remain relatively high due to a temperature inversion that often characterises the area, while during the rest of the day $b_{abs,BC}^{BB}$ gradually decrease until reaching the minimum explained above.

Comparing the first and second periods, it is interesting to note that the mean absorption coefficient for BC of the second period was significantly higher than that of the first, with mean values for $b_{abs,BC}^{FF}$ (eBC-FF) and $b_{abs,BC}^{BB}$ (eBC-BB) of 19.7 ± 15.1 $\mathrm{Mm^{-1}}$ (1.94 ± 1.49 $\mathrm{\mu g\,m^{-3}}$) and 4.5 ± 5.9 $\mathrm{Mm^{-1}}$ (0.44 ± 0.58 $\mathrm{\mu g\,m^{-3}}$), respectively, at the traffic site for the first period and 29.4 ± 24.6 $\mathrm{Mm^{-1}}$ (2.90 ± 2.43 $\mathrm{\mu g\,m^{-3}}$) and 13.3 ± 11.2 $\mathrm{Mm^{-1}}$ (1.32 ± 1.11 $\mathrm{\mu g\,m^{-3}}$) for the second period at the same station. $b_{abs,BC}^{BB}$ (eBC-BB) at the urban background site expresses the similar behaviour with a mean value equal to 5.2 ± 4.7 $\mathrm{Mm^{-1}}$ (0.52 ± 0.46 $\mathrm{\mu g\,m^{-3}}$) for the first period and to 9.6 ± 8.7 $\mathrm{Mm^{-1}}$ (0.95 ± 0.86 $\mathrm{\mu g\,m^{-3}}$) for the second period. A possible explanation for this increase can be found in the different meteorological conditions in the two periods. More specifically, the second period shows more stagnant conditions compared to the first period. This is due to more stable atmospheric conditions, characterised by recurrent thermal inversions (Figures S9 and S10) and a higher frequency of stable atmospheric classes compared to the first period (Figures S2 and S3) which have likely led to lower mixing layer height. The frequency of occurrence of stable atmospheric classes (G and F, according to the Pasquill and Guilford formulation) is 0.53 for the first period and 0.63 for the second period. In addition, the mean hourly temperature observed at the OSS station, representing the urban area, was significantly lower in the second period (2.9 °C) compared to the first period (9.4 °C). This

probably led to an increase in biomass burning for domestic heating, contributing significantly to the total eBC concentrations (see Figures 2).

On the other hand, $b^{FF}_{abs,BC}$ (eBC-FF) at the urban background site shows a different trend. Contrary to the other site and $b^{BB}_{abs,BC}$, its mean value decreased from 9.8 ± 7.1 $\mathrm{Mm^{-1}}$ (0.97 ± 0.70 $\mathrm{\mu g\,m^{-3}}$) in the first period to 7.3 ± 5.5 $\mathrm{Mm^{-1}}$ (0.72 ± 0.55 $\mathrm{\mu g\,m^{-3}}$) in the second period. The most plausible explanation for this discrepancy lies in the type of days included in the second period. Specifically, the measurements at the urban background site were taken over a period of 13 days, 7 of which coincided with holidays during the Christmas time, resulting in lower traffic volumes throughout the city. It is therefore reasonable to expect a limited contribution from traffic during this period. Furthermore, it is worth noting the impact of strong stagnation conditions that occurred in Modena between 13-14 and 18-19 January 2021 (see Figure S2 and Figure 5), verified after the end of the measurements at the urban background site.

Numerous studies have reported eBC levels across Europe. In recent years, for example, Savadkoohi et al. (2023) carried out a comprehensive comparison of eBC levels at 50 monitoring sites, including the cities of Paris, Milan, Bern and Lille for the periods 2016-2019, 2019-2021, 2015-2021 and 2017-2019 respectively. At traffic sites in Paris and Milan, the eBC-FF concentrations (95% confidence interval) were 2.17 (0.55, 5.06) $\mathrm{\mu g\,m^{-3}}$ and 2.68 (0.75, 6. 11) $\mathrm{\mu g\,m^{-3}}$, while the eBC-BB concentrations at the same two sites were 0.31 (0.00, 0.99) $\mathrm{\mu g\,m^{-3}}$ and 0.53 (0.07, 1.88) $\mathrm{\mu g\,m^{-3}}$, respectively. At urban background sites, Bern and Lille had eBC-FF concentrations of 0.78 (0.17, 1.93) $\mathrm{\mu g\,m^{-3}}$ and 0.68 (0.14, 1. 76) $\mathrm{\mu g\,m^{-3}}$, while eBC-BB concentrations were 0.28 (0.04, 0.79) $\mathrm{\mu g\,m^{-3}}$ and 0.33 (0.04, 1.01) $\mathrm{\mu g\,m^{-3}}$, respectively. In addition, Pashneva et al. (2024) reported eBC-FF and eBC-BB concentrations of 0.78 ± 0.76 $\mathrm{\mu g\,m^{-3}}$ and 0.35 ± 0.42 $\mathrm{\mu g\,m^{-3}}$, for the city of Vilnius during winter 2021-2022. On the other hand, other urban locations of Europe such as Athens, Madrid and Rome showed average total eBC values of 2.8, 2.3, and 2.9 $\mathrm{\mu g\,m^{-3}}$, respectively for the winter periods between 2015 and 2019, between 2014 and 2015, and for January and February 2017 (Liakakou et al., 2020; Becerril-Valle et al., 2017; Costabile et al., 2017a).

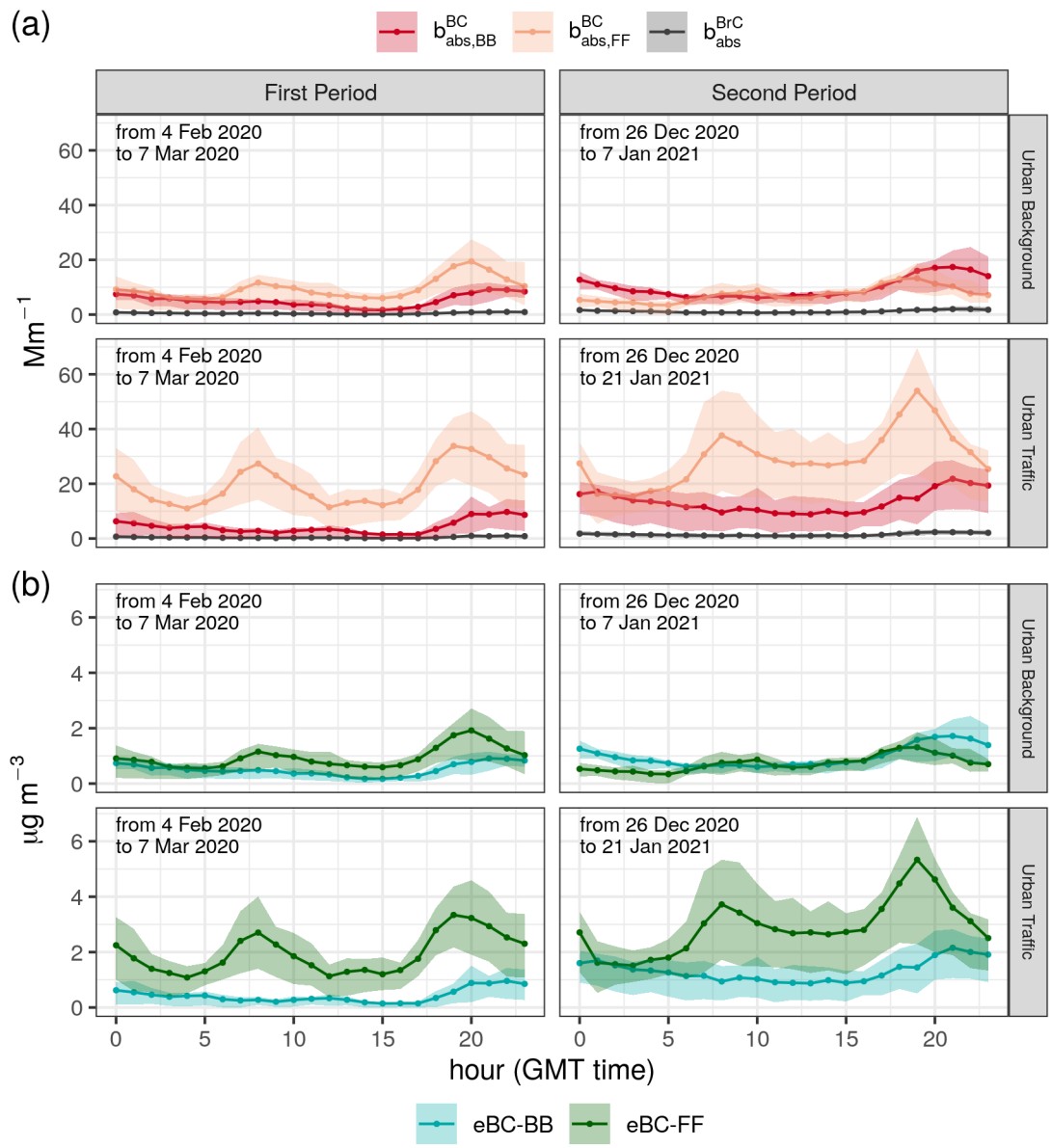

**Figure 2.** Average daily cycle of $b_{abs,BC}^{BB}$, $b_{abs,BC}^{FF}$, and $b_{abs}^{BrC}$ is shown for panel (a), while panel (b) displays eBC-BB and eBC–FF concentrations, all apportioned using the MWAA model. The solid lines represent the average absorptions or concentrations, while the shaded areas indicate the interquartile range. The top panels illustrate concentrations measured at the urban background site, while the bottom panels display concentrations measured at the traffic site.

Table 5 shows the average results of the MWAA model, providing insight into the partitioning of components (BC vs. BrC) and sources (FF vs. BB) for both the first and second periods of our investigation. Furthermore, the table shows the results of the MWAA and Aethalometer model partitioning, as well as the results of the application of the EPA PMF 5.0, using hourly absorption data and elemental concentrations, as observed in previous research carried out in the Po valley. The comparative analysis underlines the consistency of the results obtained in our study using MA200 instruments, which are on average in line with the findings of other studies carried out in the same region.

Furthermore, source apportionment results in Modena are consistent with those from other European urban areas. For instance, Savadkoohi et al. (2023) reported FF and BB partitioning at traffic sites in Stockholm, Paris, and Milan during winter, which were comparable to Modena. For the periods 2014-2019, 2016-2019, and 2019-2021, the FF and BB partitioning performed with the Aethalometer model were 80%/20%, 81%/19%, and 69%/31%, respectively. Similarly, a traffic site in Bern showed comparable source apportionment to Modena during the winter season for 2014-2018, with a FF and BB partitioning of 60%/39% (Grange et al., 2020). For urban background locations, Savadkoohi et al. (2023) also presented results for the city of Zurich, Athens, Paris and Bucharest, which were similar to those observed for the urban background location of Modena. The FF and BB partitioning during winter for these cities was 67%/31%, 59%/41%, 56%/44%, and 50%/49% for the periods 2012-2021, 2017-2019, 2016-2019, and 2014-2022, respectively. Other studies reported similar percentages for urban background locations, such as Vilnius in winter 2021-2022 (68%/31%; Pashneva et al., 2024), the surroundings of Šibenik (Croatia) in February 2019 (68%/31%; Milinković et al., 2021), the urban background of Ruhr (Germany) in winter 2013-2014 (50%/50%) and Helsinki in winter 2015-2017 (54%/46%; Helin et al., 2018).

**Table 5.** Summary table of component (BC vs. BrC) and source (FF vs. BB) apportionment based on this work and other literature studies conducted in the Po valley. Most values are reported at 880 nm, with the exception of Forello et al. (2019), which present data at 780 nm. RB, UB, and UT stand for Rural Background, Urban Background and Urban Traffic respectively.

| City | Site type | Period | Instrument | BC-FF (%) | BC-BB (%) | BrC (%) | Reference |
|------|-----------|--------|------------|-----------|-----------|---------|-----------|
| Milan | UB | Jan-Feb 2018 | AE33[a,1] | 68 | 32 | - | |
| Milan | UB | Jan-Feb 2018 | AE31[a,1] | 68 | 32 | - | |
| Milan | UB | Jan-Feb 2018 | AE33[b,1] | 72 | 28 | - | |
| Milan | UB | Jan-Feb 2018 | AE31[b,1] | 77 | 23 | - | |
| Milan | UB | Jan-Feb 2018 | in-house polar photometer[1] | 76 | 24 | - | Bernardoni et al. (2021) |
| Milan | UB | Jan-Feb 2018 | AE33[a,2] | 94[*] | | 6 | |
| Milan | UB | Jan-Feb 2018 | AE31[a,2] | 91[*] | | 9 | |
| Milan | UB | Jan-Feb 2018 | AE33[c,2] | 91[*] | | 9 | |
| Milan | UB | Jan-Feb 2018 | AE31[c,2] | 95[*] | | 5 | |
| Milan | UB | Jan-Mar 2018 | AE31 and AE51[1] | 39 | 61 | - | Mousavi et al. (2019) |
| Bareggio | RB | Jan-Mar 2018 | AE31 and AE51[1] | 64 | 36 | - | |
| Milan | UB | 21-28 Nov 2016 | in-house polar photometer[d] | 55 | 20 | - | Forello et al. (2019) |
| Milan | UB | Nov 2015-Jan 2016 | AE31[1] | 63 | 37 | - | Ferrero et al. (2018) |
| Modena | UT | Feb-Mar 2020 | MA200[2] | 80 | 18 | 2 | |
| Modena | UB | Feb-Mar 2020 | MA200[2] | 63 | 34 | 3 | This study |
| Modena | UT | Dec 2020-Jan 2021 | MA200[2] | 67 | 31 | 2 | |
| Modena | UB | Dec 2020-Jan 2021 | MA200[2] | 40 | 54 | 6 | |

[a] seven wavelength (370, 470, 520, 590, 660, 880 and 950 nm) fit with fixed multiple-scattering enhancement parameter.

[b] four wavelength (470, 520, 660 and 880 nm) fit with fixed multiple-scattering enhancement parameter.

[c] five wavelength (470, 520, 590, 660 and 880 nm) fit with fixed multiple-scattering enhancement parameter.

[d] Source apportionment was conducted using EPA PMF 5.0, incorporating hourly absorption data at four distinct wavelengths (405, 532, 635 780 nm), in combination with hourly elemental concentrations from samples gathered and analysed on the same filters. The outcomes of this analysis revealed that the third factor in the apportionment could be attributed to sulphate, constituting approximately 20% of the overall composition. The fourth factor, on the other hand, was primarily composed of resuspended dust, accounting for approximately 4% of the total, with other factors of lesser significance contributing approximately 1%.

[1] Aethalomter model.

[2] MWAA model.

[*] Total BC, represents the cumulative sum of BC-FF and BC-BB.

## 4.2 Modelling results

### 4.2.1 Meteorology

Using the GRAMM-GRAL modelling system, a comprehensive catalogue of meso- and micro-scale flow patterns was calculated. Following the methodology outlined in Section 2.4, a series of hourly wind fields were generated to drive the Lagrangian dispersion in Modena. Whenever possible, meteorological observations from the stations shown in Figure 1 were used. First, the wind fields were computed independently of any observation, following the discretisation procedure described in Section 2.4. Subsequently, the wind fields were selected to closely match the actual conditions. It is important to acknowledge that

although a given wind field from the catalogue may not be perfectly consistent with all stations simultaneously, this study relies exclusively on data from stations located in the urban area of Modena to construct the catalogue. This is due to the unavailability of measurements from other locations within the GRAMM domain and within a range of 50-60 km. Furthermore, considering that the area of interest is almost flat, we expect that this setup will not significantly affect the representation of the flow fields in the area of interest. Figure 3 shows the comparison between modelled and measured hourly wind speed for the

available urban stations for the period of interest. Panel (a) represents the first period, while panel (b) shows the wind speed comparison for the second period. Despite some notable underestimations of the observed wind speed on 5, 10, 26, 27 and 28 February 2020, due to the wind category discretisation process, GRAMM-GRAL generally reproduced the trend quite well for both periods. For the first period the wind speed is generally underestimated, with MB between -0.44 $\mathrm{m\,s^{-1}}$ and -0.27 $\mathrm{m\,s^{-1}}$ at CMP and OSS respectively (corresponding to -34% and -9% of the NMB), while at DEX the NMB is approximately zero

and the MB is 0.02 $\mathrm{m\,s^{-1}}$. Similarly, during the second period, the wind speed is generally underestimated at CMP and POL, with MB values of -0.28 $\mathrm{m\,s^{-1}}$ and -0.03 $\mathrm{m\,s^{-1}}$, respectively, while it is overestimated at OSS with an MB of 0.31 $\mathrm{m\,s^{-1}}$. The corresponding NMB values are -37%, -2%, and +20%, respectively. The RMSE of the simulated wind speed by GRAMM-GRAL is 0.88 $\mathrm{m\,s^{-1}}$, 0.72 $\mathrm{m\,s^{-1}}$ and 1.16 $\mathrm{m\,s^{-1}}$ for CMP, OSS, and DEX during the first period, and 0.60 $\mathrm{m\,s^{-1}}$, 0.83 $\mathrm{m\,s^{-1}}$ and 0.89 $\mathrm{m\,s^{-1}}$ for CMP, POL, and OSS during the second period. These RMSE values are in line with the recommended

statistical benchmark suggested by the EEA guidelines for meteorological wind field assessment (EEA, 2011), which suggests RMSE values of less than 2 $\mathrm{m\,s^{-1}}$. Other statistical indices, including FAC2, NMSE and r, are reported in Table 6 and the related scores are consistent with similar studies conducted in the same area during previous simulation years (Veratti et al., 2020, 2021). To complement the wind speed analysis, Figure S4 displays wind roses comparing the modelled and observed wind speed and direction for the same stations discussed above. The wind roses confirm that the modelling system effectively

reproduces many of the observed features, with the observed winds being generally well captured by GRAMM-GRAL, especially during the second period.

Further analysis was carried out on the height of the planetary boundary layer (PBL), a key driver of atmospheric dispersion. Although direct measurements of the PBL height were not available for Modena, we made both quantitative and qualitative comparisons with observations taken in rural areas at the S. Pietro Capofiume (SPC) station, approximately 50 km east of

580 Modena (see Figure 1 and Table 1). Specifically, we estimated the PBL height using the Richardson number (Ri) derived from sounding data at 00:00 and 12:00 GMT, using 0.25 as the critical value for identifying turbulent conditions (Lyons et al., 1964;

Galperin et al., 2007; Grachev et al., 2013). These estimates were then compared with the PBL height simulated by NINFA, which also uses the Richardson number in its calculations (Troen and Mahrt, 1986). Figures S5 and S6 in the supplementary material provide an overview of this comparison for two periods: 15 February to 7 March 2020 and 26 December 2020 to 21 January 2021. For the first period, data was only available at 00:00 GMT, while for the second, data was also available at 12:00 GMT. The results show that between 15 February to 7 March 2020, NINFA generally underestimates the sounding estimates, with a MB of -100 m (-52%), resulting in a shallower mixing layer compared to the measurements. However, during the second period, the PBL height was better reproduced, with a limited MB of -47 m (-27%) at 00:00 GMT and -38 m (-12%) at 12:00 GMT, showing a robust performance in simulating the vertical structure of the atmosphere. Despite the more pronounced underestimation of the first period, NINFA showed similar performance to other meteorological models applied in the Po valley and other locations in Italy in reproducing the PBL height derived from soundings (Ferrero et al., 2011; Avolio et al., 2017).

For a qualitative comparison, the PBL height simulated by GRAL over the urban area of Modena is also included in Figures S5 and S6. Although a quantitative analysis between GRAL and soundings is not possible due to the fact that the measured area is outside the GRAL domain and various factors may cause differences in the vertical turbulence profile between urban and rural areas (e.g. urban heat island effects and anthropogenic heat sources), this comparison serves as a basis for hypothesising whether GRAL can realistically reproduce the PBL height during sounding time.

The results of the GRAL simulations show that during the first period the PBL height is in agreement with the sounding data on most days (16 out of 22). However, on 6 of the 22 days, GRAL significantly overestimates the measurements at 00:00 GMT, with values up to 800 m, which corresponds to the domain top internally set in the model code. During the second period, the PBL height modelled by GRAL generally matches the soundings data at both 00:00 GMT and 12:00 GMT, except for nine days when there is a significant overestimation (up to four times the observations) at 12:00 GMT. A more detailed analysis of these episodes shows that during the night (00:00 GMT), when stable or very stable atmospheric conditions are imposed on GRAL, the PBL height tends to match the values observed at SPC. Conversely, under neutral conditions, the simulated PBL height by GRAL overestimates the observations by a factor of three to six. This overestimation under neutral conditions is also evident at 12:00 GMT during the second period, with 6 out of 9 episodes occurring under these conditions.

Despite these challenges faced by GRAL, a common limitation for urban air quality models (Sokhi et al., 2022), the simulated PBL height appears realistic on most days. In addition, it is important to note that the limited number of observational data points prevents further conclusions being drawn for other times of the day.

**Table 6.** Statistical analysis of hourly wind speed computed at urban meteorological stations using the GRAMM-GRAL modelling system.

| Station | Period | MB $(\mathrm{m\,s^{-1}})$ | NMB (%) | RMSE $(\mathrm{m\,s^{-1}})$ | NMSE | FAC2 | r |
|---------|--------------|-------|----|------|------|------|------|
| CMP | first period | -0.44 | 34 | 0.88 | 0.27 | 0.67 | 0.71 |
| DEX | first period | 0.02 | 0 | 1.16 | 0.72 | 0.73 | 0.77 |
| OSS | first period | -0.27 | 9 | 0.72 | 0.07 | 0.97 | 0.95 |
| CMP | second period | -0.28 | 37 | 0.60 | 0.98 | 0.42 | 0.55 |
| OSS | second period | 0.31 | 20 | 0.83 | 0.23 | 0.76 | 0.75 |
| POL | second period | -0.03 | 2 | 0.89 | 0.23 | 0.72 | 0.61 |

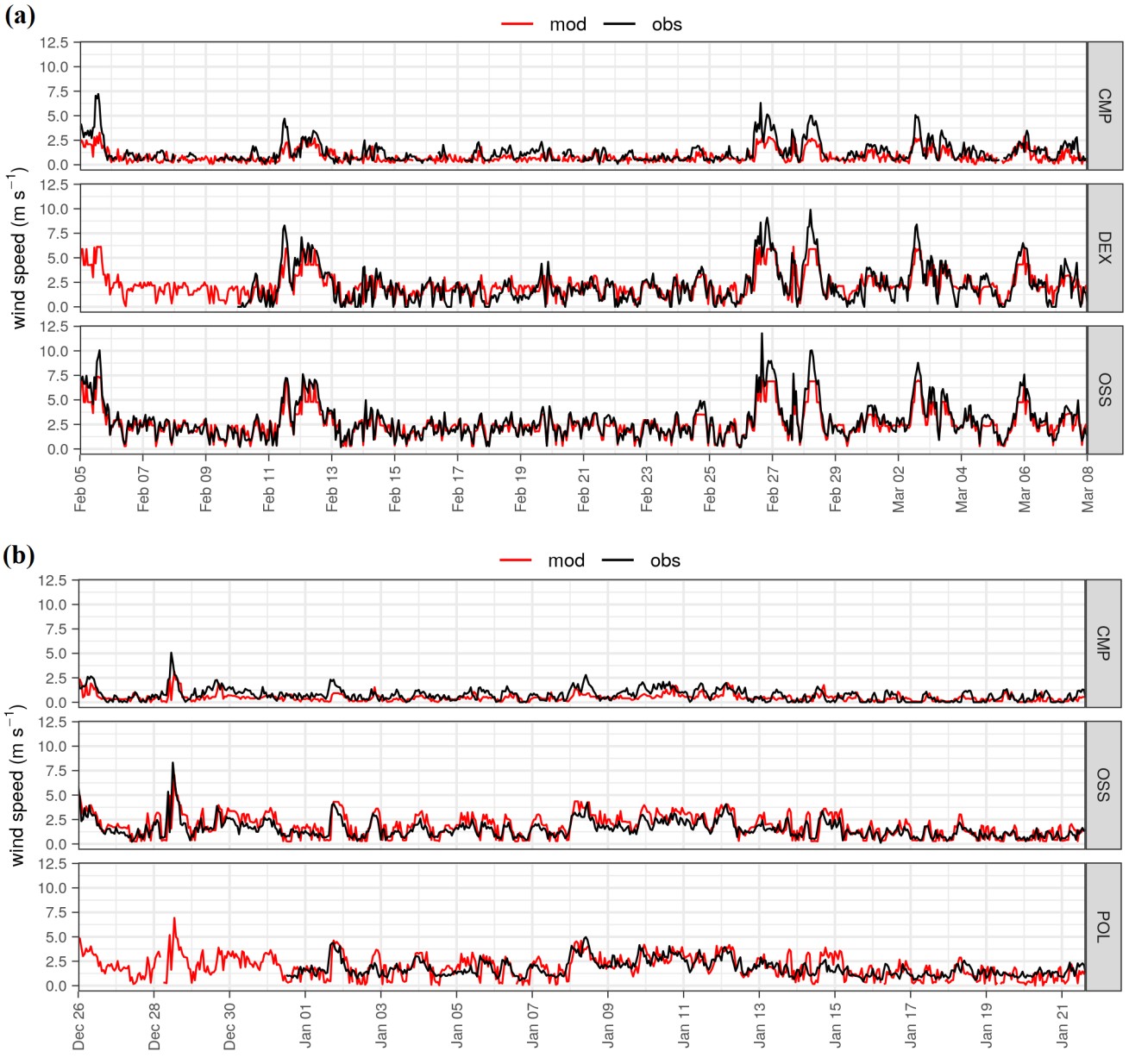

**Figure 3.** Hourly time series of measured wind speed (black) at four urban meteorological stations in Modena, alongside micro-scale modelled values by GRAMM-GRAL (red) for two periods: (a) 4 February to 7 March 2020, and (b) 26 December 2020 to 21 January 2021.

### 4.2.2 Comparative analysis of modelled BC concentrations using a hybrid Eulerian-Lagrangian modelling system in Modena

The background BC concentrations derived from NINFA in the urban area of Modena were integrated with the BC concentrations simulated by the GRAMM-GRAL modelling system, which specifically considers emissions only at the urban scale. The scenarios considered for the evaluation include TRF-exh-1 for traffic exhaust emissions and TRF-nexh-1 for non-exhaust emissions. In the case of resuspension and biomass burning, which are characterised by a high level of uncertainty, the average concentrations calculated from all simulated scenarios were taken into account. For NINFA, the scenarios TRF-bck-1 and DMH-bck-1 were considered. This specific emission configuration will be referred to as the base case scenario.

To evaluate the effectiveness of the hybrid modelling system in various urban settings, modelled concentrations at station locations were compared with observed eBC levels from MA200 devices. Figures 4 and 5 show the time series comparing measured (eBC) and modelled (BC) concentrations, divided into their contributors (biomass burning, fossil fuels and the sum of the two) for the first and second periods, respectively. The analysis reveals a notably robust performance of the hybrid system in modelling concentrations during the first simulated period (5 February - 8 March, 2020), despite BB being generally overestimated at the traffic site. The bias of total modelled BC concentrations is very low at both sites: MB is -0.12 $\mu g\, m^{-3}$ for urban traffic (corresponding to -5% of NMB) and 0.03 $\mu g\, m^{-3}$ for urban background (corresponding to +2%). The Pearson's correlation coefficient between modelled and observed concentrations is 0.62 at the traffic site and 0.51 at the urban background station. In addition, the modelling system generally captured the daily peaks correctly, even under different meteorological conditions. In particular, several relatively high wind speed episodes for typical Modena meteorological conditions occurred during this period, such as on 5, 11, 12, 26, 28 February, and 2, 5 March 2020, with speeds reaching 11.2 $m\, s^{-1}$. The modelling system showed adaptability in simulating both calm and windy periods. Conversely, the performance of the model in reproducing observations decreased during the second period (26 December 2020 - 21 January 2021), as observed eBC concentrations experienced significantly higher values than in the first period. The Pearson's correlation coefficients between modelled and observed concentrations are 0.34 and 0.38 at the traffic and background sites, respectively. The corresponding MB are -0.92 $\mu g\, m^{-3}$ (corresponding to -22% of NMB) and +0.25 $\mu g\, m^{-3}$ (corresponding to +15% of NMB) at the same two stations. Favourable meteorological conditions for the accumulation of pollutants were observed both in Modena and in the surrounding areas. These conditions were mainly characterised by high pressure systems and persistent thermal inversions in the lower atmospheric layers, resulting in a likely shallow mixing layer height.

This meteorological pattern contributed to the elevated eBC concentrations observed at both sites, particularly at the traffic station where levels reached up to 18 $\mu g\, m^{-3}$ (Figure 5). In these circumstances, the hybrid system struggled to reproduce the observed concentration pattern. A possible explanation is the difficulty of the GRAL model in accurately simulating the PBL height over the urban area. As shown in Section 4.2.1, while GRAL generally agreed with the observations at 00:00 GMT and 12:00 GMT, it failed to produce a realistic PBL height during certain sporadic episodes, particularly under neutral conditions at night. It can therefore be hypothesised that GRAL has difficulty in simulating the very stagnant meteorological conditions that can occur in the Po valley. Although this limitation exists, similar challenges have been documented in the literature for

various other models and regions, as noted by Fay and Neunhäuserer (2006); Saide et al. (2011); Tominaga and Stathopoulos (2016); Travis et al. (2022). Continuous measurements of the PBL height in Modena would be necessary to further analyse the

accuracy of GRAL during the day with respect to this parameter.

Looking more closely at specific days, namely 3-4, 13-14 and 18-19 January 2021, significant peaks in BC levels were observed. At the traffic site, the peak concentrations on these days were 14.1 $\mu gm^{-3}$, 18.5 $\mu gm^{-3}$ and 16.3 $\mu g\,m^{-3}$, respectively. Analysing the episodes on 13-14 and 18-19 January 2021, while direct BC data at the background site are not available for this period, the total $PM_{10}$ and $PM_{2.5}$ concentrations recorded at the same two locations (Figure S7) confirm the presence

of meteorological and emission conditions conducive to pollutant accumulation, with peak values of 105 $\mu g\,m^{-3}$ for $PM_{10}$ and 66 $\mu g\,m^{-3}$ for $PM_{2.5}$. Figure S2 shows a significant increase in atmospheric pressure at OSS, suggesting the establishment of a high pressure system in the area. This atmospheric scenario was accompanied by calm and stable meteorological conditions, with an average wind speed of less than 2.5 $m\,s^{-1}$ (Figure 3, panel b). Furthermore, these conditions contributed to the formation of temperature inversions not only during the night but also during the day, as shown in Figures S9 and S10.

Taken together, these conditions probably facilitated the accumulation of pollutants near the ground, thereby limiting both their vertical and horizontal dispersion. Comparable meteorological conditions were also observed on 16 January 2021. However, unlike the previously analysed days, 16 January fell on a Saturday, which is typically associated with lower anthropogenic emissions from traffic and industrial activities compared to weekdays. Consequently, the BC and PM concentrations on this particular day were lower than those recorded on 13-14 and 18-19 January 2021.

On the other hand, the interpretation of the episode recorded on 3 and 4 January 2021 is somewhat more challenging. On these days, the total $PM_{10}$ and $PM_{2.5}$ concentrations were typical for the period, with no significant peaks (Figure S7). In particular, daily mean $PM_{10}$ concentrations were around 30 $\mu gm^{-3}$ and 25 $\mu gm^{-3}$ at the traffic and background stations respectively, while daily mean $PM_{2.5}$ concentrations were around 14 $\mu gm^{-3}$ at the background station. Vertical temperature profiles derived from soundings at 00:00 and 12:00 GMT at SPC (Figures S8) indicated only a shallow inversion layer near the ground, in con-

trast to the strong inversion layer observed on 13-14 and 18-19 January. In addition, other meteorological variables recorded at OSS did not suggest conditions conducive to a particular episode. It is possible that during the hours of peak concentration (18:00 GMT on 3 January and 08:00 GMT on 4 January), the inversion layer, combined with a temporary reduction in wind speed, may have decreased for only a limited number of hours. This may have exacerbated the eBC concentrations recorded at high temporal frequency, while the daily averages of $PM_{10}$ and $PM_{2.5}$ were moderately affected by these short episodes. Fur-

thermore, the presence of sporadic high emission sources near the traffic station, such as high emitting vehicles passing nearby or idling in nearby car parks, could have caused the peak on 4 January. Indeed, this peak was not observed at the background site and was dominated by fossil fuel contributions, suggesting the possible presence of local sources.

Despite the decrease in the performance of the hybrid modelling system compared to the first period, the metrics of FB, NMSE, FAC2 and NAD closely match with the results reported by Lugon et al. (2021) when simulating BC concentrations in a

suburban street network of Paris using the Street-in-Grid model. Furthermore, the corresponding metrics of FB, NMSE, FAC2 and NAD in this study for both periods, reported in Table 7 together with the previously mentioned indices, largely meet the

**Table 7.** Statistical analysis of hourly total BC concentrations at urban traffic and urban background sites using the hybrid modelling system.

| Station | Period | MB ($\mu$g m$^{-3}$) | NMB (%) | RMSE ($\mu$g m$^{-3}$) | NMSE | FAC2 | r | FB | NAD |
|---------|--------|------|-----|------|------|------|------|------|------|
| Urban Traffic | first period | -0.12 | -5 | 1.38 | 0.34 | 0.80 | 0.62 | 0.05 | 0.02 |
| Urban background | first period | 0.03 | 2 | 1.02 | 0.34 | 0.67 | 0.51 | 0.05 | 0.02 |
| Urban Traffic | second period | -0.92 | -22 | 2.89 | 0.60 | 0.69 | 0.34 | 0.25 | 0.13 |
| Urban background | second period | 0.25 | 15 | 1.13 | 0.35 | 0.72 | 0.38 | -0.13 | 0.06 |

acceptance criteria established by Hanna and Chang (2012) for urban dispersion modelling, where the benchmark values are defined as follows:

- FAC2 > 0.30

- NAD < 0.50

- |FB| < 0.67

- NMSE < 6

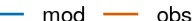

**Figure 4.** Hourly time series of observed (eBC) and simulated (BC) concentrations at urban traffic and background sites for the period from 4 February to 7 March 2020. Both simulated and observed concentrations are presented as individual contributions from fossil fuel (FF), biomass burning (BB), and their combined total (Total).

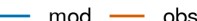

**Figure 5.** Hourly time series of observed (eBC) and simulated (BC) concentrations at urban traffic and background sites for the period from 26 December 2020 to 21 January 2021. Both simulated and observed concentrations are presented as individual contributions from fossil fuel (FF), biomass burning (BB), and their combined total (Total). Please note the difference in the scale on the y-axis between background and traffic stations.

The MWAA model, applied to MA200 measurements with additional hypotheses about $\alpha$ for different sources, enabled to dishern the contributions of different sectors to observed eBC concentrations (see Section 2.1). This approach allowed for the assessment of the hybrid system's performance in reproducing concentrations of two distinct sources: fossil fuel and biomass burning. Figures 6 and 7 present the diurnal patterns of observed and modelled concentrations, respectively for the first and second simulated periods, for total BC (eBC/BC, panel a), BC from fossil fuel (eBC-FF/BC-FF, panel b), and BC from biomass burning (eBC-BB/BC-BB, panel c). In addition, in order to assess the influence of emission uncertainties on the model outputs and explore potential outcome ranges, the figures for BC-FF and BC-BB also include the uncertainty range resulting from the emission scenarios described in Section 2.5. Error bars indicate the minimum and maximum average concentrations derived from different scenarios. The points correspond to the average concentrations of the base case scenario. Additionally, to provide additional details, blue shaded areas depict the interquartile range derived from the base case scenario, while red shaded areas represent the interquartile range of observed concentrations.

Due to a limitation in the MWAA model, it is not possible to separate the influence of non-exhaust and resuspension from the total eBC concentrations without additional experimental data. Therefore, where not explicitly stated, the modelled BC-FF concentrations in this section include the combined contribution of fossil fuels, traffic non-exhaust and traffic resuspension.

During the first period (as shown in Figure 6, panel a), it is noteworthy that despite a slight underestimation of the measured concentrations occurring between 03:00 and 06:00 GMT and a slight overestimation observed between 18:00 and 21:00 GMT both at the urban background site, the simulation shows a good agreement with measurements at both locations. Looking at the specifics of the eBC-FF/BC-FF concentrations (Figure 6, panel b), the base case scenario proves to be generally effective in representing the average trends, despite a slight underestimation between 18:00 and 21:00 GMT at both sites during the second peak of daily rush hours. This discrepancy could probably be attributed to an inherent underestimation of traffic flows during this specific time period, a trend that is consistent with the results of the second period and with the findings of Veratti et al. (2021), who performed a comprehensive analysis of the same area with respect to nitrogen oxides concentrations. In order to test this hypothesis in depth, we compared the measured traffic flows from 45 induction loop sensors located at traffic lights in the vicinity of the two monitoring stations with the traffic flows simulated by PTV VISUM on the reference roads around these stations. The results showed that on 36 out of the 57 simulation days, the traffic flows between 18:00 and 21:00 GMT were underestimated, with NMB of up to -18% for roads near the urban traffic station and up to -13% for roads surrounding the urban background site. Despite this underestimation, the overall agreement of model results with observations further confirms the effectiveness of a bottom-up approach in accurately representing traffic emissions.

Focusing on the contribution of biomass burning to total BC (Figure 6, panel c), it is interesting to note that during the first half of the day, particularly between 03:00 and 11:00 GMT, the observations are close to the lower bound indicated by the error bars. This suggests that the collective average of all scenarios tends to slightly overestimate the observed trend during this period. Conversely, the experimental eBC-BB pattern could also be underestimated, possibly due to measurement uncertainties associated with the MA200 instruments, as reported in recent studies (Alas et al., 2020; Li et al., 2021). More specifically, the observations between 03:00 and 11:00 GMT show a stronger agreement with the results generated by the DMH-2 scenario at the urban background location and by the DMH-4 scenario at the urban traffic location. Conversely, the peak in eBC-

BB concentrations occurring after 19:00 GMT is remarkably well represented by the average concentration derived from all modelled scenarios. These results lead to two plausible interpretations. The first suggests that the hourly modulation profile used to distribute the emissions over the day takes into account a morning peak that may not be prevalent on average among the sources within Modena (see Figure S12 in the Supplement for the emission daily modulation profile used in this study). The same hypothesis suggests that the distribution of emissions among the different districts of the city could possibly be an intermediate situation between those presented in this study (i.e. DMH-1, DMH-2, DMH-3, DMH-4 and DMH-5 scenarios). The second alternative explanation suggests that the emission allocation is a mixture of those represented by scenarios DMH-2 and DMH-4 and that the hourly modulation profile underestimates the emission peak during the night hours when the observations express the highest concentrations.

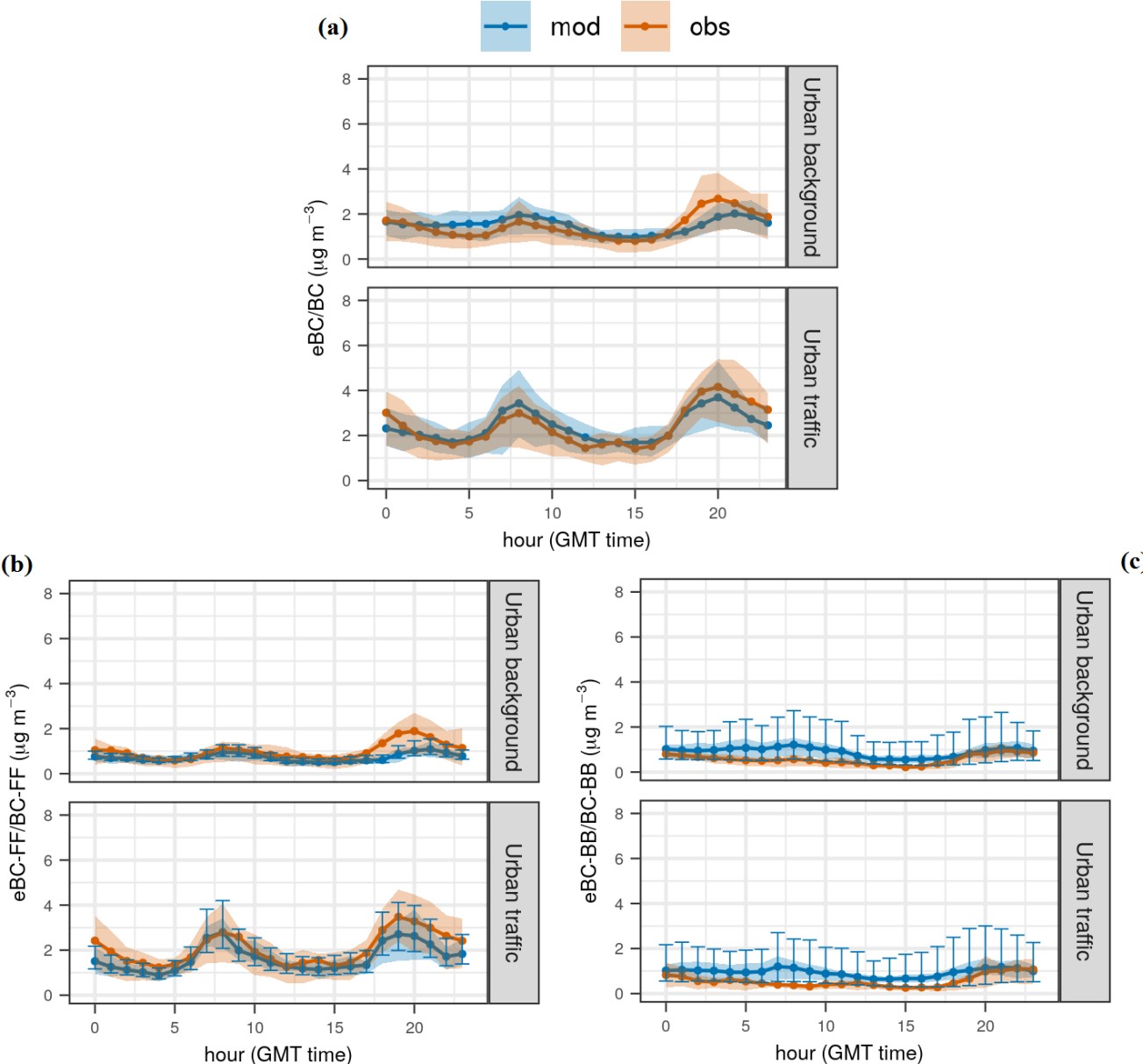

**Figure 6.** Comparative analysis of modelled (BC, blue) and observed (eBC, red) mean daily cycles during the first period for total BC concentrations (Panel a), Fossil Fuel-derived BC (Panel b), and Biomass Burning-derived BC (Panel c). The shaded regions in all three panels depict the interquartile range encompassing observations (red) and model outcomes (blue) for the base scenario. Error bars in Panels (b) and (c) delineate the minimum and maximum average concentrations derived from the simulated scenarios in this study. Dots correspond to average concentrations from the base case scenario.

When analysing the second period, a contrasting behaviour in the modelled concentrations is observed between the traffic and background sites (Figure 7, panel a). At the urban background station, the modelled concentrations are generally overestimated during most of the night and morning hours. However, the magnitude of the night peak is well captured, albeit with a one hour delay, probably due to the emission modulation profiles imposed on GRAL during the Christmas holidays for both FF and BB (Figure S12), which do not seem to fully represent the observed pattern. On the other hand, the measured concentrations at the traffic site are consistently underestimated during every hour of the day, although the hourly trend is well represented by the hybrid modelling system.

Despite the divergent results observed at the two stations, it's important to bear in mind that the concentrations represented at the two stations refer to two different periods: measurements were carried out from 26 December 2020 to 7 January 2021 at the urban background site and from 26 December 2020 to 21 January 2021 at the urban traffic site. Therefore, these data sets are not directly comparable (see previous explanations in this section). When analyses are restricted to periods common to both sites, model concentrations at the traffic site show similar behaviour to the urban background site. The hybrid modelling system reports an MB of 0.42 $\mu g\,m^{-3}$, corresponding to +16% of the NMB, while the Pearson's correlation coefficient between modelled and observed values is 0.34, which is very similar to the statistics reported at the background site (Table 7). This overestimation is mainly due to three different episodes occurring between 27-28 December 2020, 4-5 January 2021 and 6 January 2021. The first and last episodes are evident at both sites, with both FF and BB affected by an overestimation of the same magnitude in relative terms. This suggests that the local meteorology was not well captured by GRAL, probably resulting in a lower PBL height than the actual values. On the other hand, the episode between 4 and 5 January 2021 is dominated by an overestimation of the BB component only at both sites. In this case, the concept of heating degree days used to assign the daily BB emissions to each simulated day may not have accurately represented the actual emissions for that specific day.

Looking more closely at the contributions of individual sources to the total eBC/BC concentrations at the urban background site, a detailed examination shows that the overestimation in the morning peak is due to an overestimation of biomass burning emissions (Figure 7, panel c), which is consistent with the results of the first period. Conversely, the night peak is well captured by the average concentration derived from all biomass burning scenarios considered. The modelled BC-FF concentrations, although showing very limited variability throughout the day, are in good agreement with the measurements, suggesting that a precise modulation of traffic and other fossil fuel related sources for holidays (which predominantly characterise most days of the campaign at the background site) could have been applied.

At the traffic site, biomass burning concentrations are well simulated by the modelling system, with observations closely matching the averaged concentrations from all simulated scenarios for most hours, except for the morning peak, confirming the previously developed hypothesis. In more detail, the results of the second period suggest a more plausible scenario where a single daily peak occurs at night between 20:00 and 22:00 GMT. Moreover, the daily modulation profile based on the concept of heating degree days seems to be quite effective in consistently reproducing biomass emissions for domestic heating under different winter temperatures, despite the overestimation reported between 4 and 5 January 2021. The modelled BC concentrations originating from fossil fuels at the traffic site show a tendency to converge towards the upper limits of the error bars corresponding to the upper range of the BC/PM speciation factors recommended by the EEA, i.e. by the combination of

the TRF-exh-3, TRF-nexh-3 and TRF-bck-3 scenarios. While this result might suggest that the upper range of the speciation factors might reflect the composition of the vehicle fleet in Modena, we believe that the source of this underestimation is mainly related to the difficulties encountered by the modelling system in simulating specific meteorological events and the presence of emitting sources that pose challenges to the simulation.

As discussed earlier, instead of a consistent underestimation of the observed trend throughout the simulated period due to low speciation factors, distinct episodes of significant underestimation can be identified (see Figure 5). These episodes were characterised by a robust meteorological stagnation (such as the events of 13-14 June and 18-19 June 2021), which GRAL is unlikely to have captured. To support the hypothesis of inaccuracies in the simulation of the pronounced atmospheric stability observed on 13-14 and on 18-19 January 2021, Figure S11 in the supplementary material provides an overview of the hourly simulated PBL height by GRAL in Modena, a critical variable for vertical mixing and pollutant dispersion. Although a direct comparison with measured values within Modena is not possible due to a lack of measurements, it is evident that the diurnal PBL height for the mentioned critical days (13-14 and 18-19 January 2021) is close to or even higher than that of the previous days, without any particular trend towards lower values. Consequently, the vertical dispersion of BC emitted at the city scale on these days was somewhat overestimated. Despite these inherent limitations, the results obtained from both the first and second periods show that the method used to estimate fossil fuel emissions, especially those from traffic, has proved its effectiveness in accurately reproducing the observed trend.

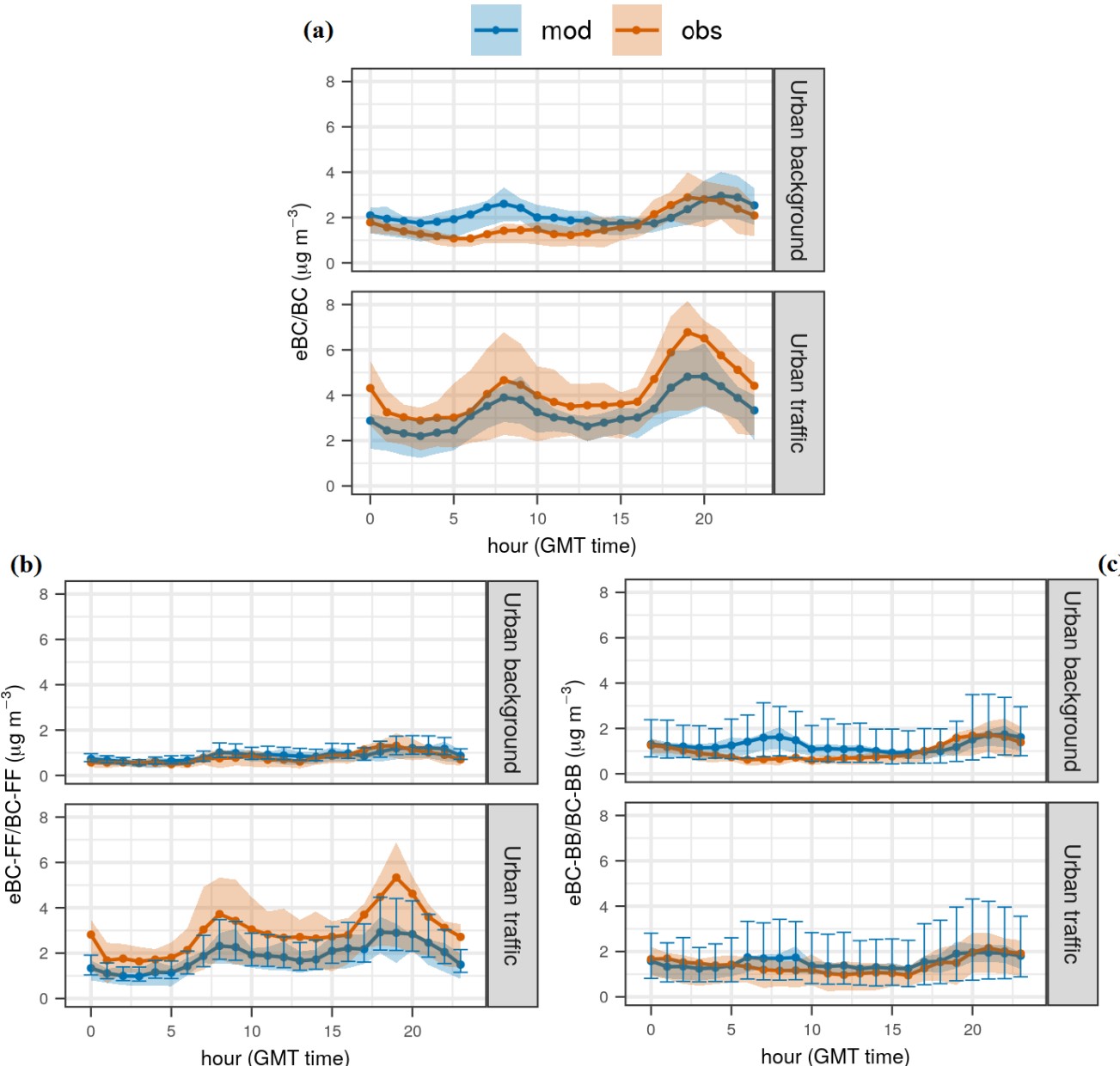

**Figure 7.** Comparative analysis of modelled (BC, blue) and observed (eBC, red) mean daily cycles during the second period for total BC concentrations (Panel a), Fossil Fuel-derived BC (Panel b), and Biomass Burning-derived BC (Panel c). The shaded regions in all three panels depict the interquartile range encompassing observations (red) and model outcomes (blue). Error bars in Panels (b) and (c) delineate the minimum and maximum average concentrations derived from the simulated scenarios in this study. Dots correspond to average concentrations from the base case scenario.

A limitation of our study is that the modelled BC concentrations are treated as inert and no external or internal mixing coatings were simulated by the hybrid modelling system. While the simulation of coating absorption is particularly challenging (Curci et al., 2019), especially for Lagrangian models (Doran et al., 2008; Holmes and Morawska, 2006), omitting this aspect could introduce potential uncertainties during model validation when assuming reference MAE values for aethalometer measurements. In fact, different mixing states can lead to an enhancement of the BC absorption efficiency ($E_{abs}$) compared to the theoretical pure form of BC. Studies carried out at different locations around the world have reported a wide range of $E_{abs}$ values, from almost negligible (Cappa et al., 2012) to 2.4 (Peng et al., 2016), with most falling in the range of 1.2-1.6 (Moffet and Prather, 2009; Bond et al., 2006; Liu et al., 2017; Schwarz et al., 2008). To account for the uncertainty associated with MAE values when comparing modelled and measured concentrations, we investigated the impact of varying MAE. In particular, although we found that about half of the modelled BC concentrations in Modena originate from the city itself (as detailed in section 4.2.3), where we assume a negligible coating, especially when analysing concentrations close to emission sources, it is reasonable to expect that regional BC emissions originating from outside Modena may undergo ageing processes and thus experience absorption enhancement. Taking these factors into account, we examined the effects of adjusting the MAE within a range of ± 20%, which represents half of the average absorption enhancement range (1.2-1.6) documented in numerous studies in the literature. Figure S13 in the supplementary material shows the linear regression between modelled and measured BC concentrations, using data from both simulated periods and the reference MAE values presented in Table S1. Setting the intercept to zero yields slope values of 0.83 and 0.65 when evaluating total BC concentrations at background and traffic sites. At the same sites, the RMSE of the regression model is 1.0 and 1.7, respectively. Focusing only on the contribution of biomass burning, the slope of the regression line is 0.98 and 0.77 at background and traffic sites, respectively, with corresponding RMSE values of 0.7 and 1.0. For the fossil fuel contribution, the slope of the regression line is 0.49 and 0.56 and the RMSE is 0.63 and 1.2 at background and traffic sites, respectively. By increasing the MAE by 20% compared to the values reported in Table S1, both the slope of the regression line and the RMSE of the linear model increase by about 17%. Concurrently, the NMB between modelled and observed concentrations shifts from +7% to -11% at the urban background site and from -0.16% to -30% at the traffic site when assessing total BC concentrations. Conversely, a 20% reduction in the MAE results in a reduction of approximately 25% in both the slope of the regression line and the RMSE of the linear model. This variation in MAE results in an NMB of +34% at the urban background site and +5% at the traffic site when analysing total BC concentrations. The outcomes of the linear regression, together with the NMB values for the three scenarios analysed, are presented in Table S3 in the supplementary material. These results include assessments of total BC concentrations as well as contributions from biomass burning and fossil fuels.

### 4.2.3 Dispersion modelling based source apportionment

Once the robustness of the hybrid modelling system in reproducing the observed trends was verified, the second phase of the study employed the same system to identify potential sources influencing urban BC concentrations in Modena. The design of the modelling framework allows a direct distinction between contributions originating from within the city (labelled as City in the Figures 8 and 9) and from sources located outside the urban area (labelled as bck in the Figures 8 and 9). This

distinction is achieved through adjustments made to the CHIMERE code, enabling real-time apportionment for non-reactive BC. In addition, the ability of the Lagrangian model GRAL to store concentration fields for selected sources, allow direct apportionment of different urban emission sectors.

Figure 8 shows the average spatial contributions from each simulated source, which were calculated by analysing hourly maps from both the first and second simulated periods. In panel (a) of this figure, the upper facets show the average contributions of BB, FF, traffic non-exhaust and resuspension emissions from urban sources (labelled as City). Meanwhile, the lower facets show the average contributions from the same sources, but originating from areas outside the region of interest and transported over longer distances (labelled as bck).

When focusing on the contributions originating from within the city, these maps underscore the significant impact of BB on the overall levels of BC, as evident in the top-left facet of panel (a). Indeed, even though BC emissions are potentially absent or extremely low within the historical city centre, concentrations tend to spike in the surrounding area, reaching up to $2 \, \mu g \, m^{-3}$ only from this contribution. This can be ascribed to the increasing density and height of buildings as one moves from rural areas into the historical centre. These factors give rise to intricate urban canyons that foster atmospheric stagnation, thereby
hindering the dispersion and dilution of pollutants. Regarding FF contributions from urban sources (centre top-left facet of panel a), it is evident that traffic is the primary source of FF combustion, as indicated by the concentration peaks of BC. These peaks exhibit noticeable high spatial gradients, particularly prominent in streets characterised by heavy traffic, such as the urban ring road encircling the urban area, the small section of highway in the bottom-left corner of the domain and several other bustling urban streets. Furthermore, non-exhaust emissions and resuspension (respectively centre top-right and top-right
facets of panel (a) also make a notable contribution to total BC concentrations, even if their absolute values are lower compared to those of BB and FF, especially in the vicinity of the aforementioned heavily trafficked streets.

In the lower facets of panel (a), the contributions from long-range transport are displayed. Although these maps show more uniform concentration patterns when compared with those at the city scale, BB concentrations remain the dominant contributor among the four source categories considered here. This underscores that emissions outside the urban centre also exert a
noteworthy influence on urban BC concentrations. On the other hand, panel (b) of Figure 8 represents the sum of each single contribution from panel (a). This plot displays the capability of the hybrid modelling system in simulating urban concentration gradients originating from different source locations and it also underscores the ability of the Lagrangian modelling suite to reconstruct the impact of urban canyons, which play a crucial role in the accumulation of pollutants in the most critical areas of the city centre.

Based on the results presented in Figure 8 and considering a receptor location equivalent in scale to the GRAL simulation domain, the average concentration over all cells in this area was calculated for all emission sources analysed here. The reference value reported for each contribution represents the average of all simulated scenarios for the respective contribution (see Table 2 and Table 3), and the associated uncertainty is the resulting standard deviation from these scenarios. A breakdown of the local urban contribution shows that BB accounts for 35% ± 15%, FF for 9% ± 4%, non-exhaust emissions for 4% ± 2% and
resuspension for 4% ± 3%. It's also worth noting that FF is mainly attributed to traffic sources, with other FF sources accounting for less than 1%. In contrast, according to the Arpae local emission inventory, total BB emissions are roughly evenly distributed

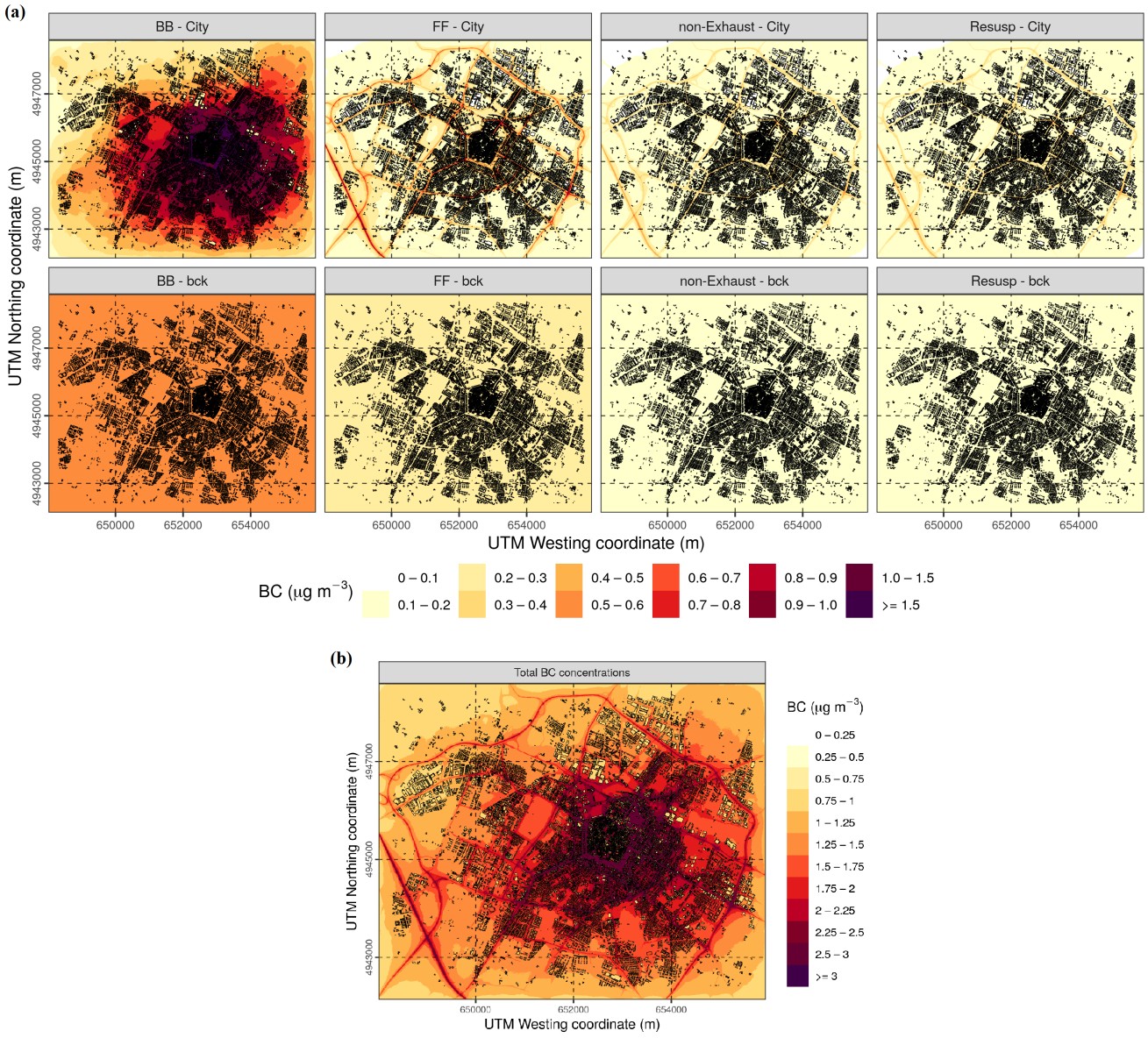

**Figure 8.** Modelled breakdown of BC contributions, differentiating between fossil fuels (FF), biomass burning (BB), traffic non-exhaust emissions and resuspension (panel a). The upper facets represent the BC contributions from local sources within the city (City), while the lower facets depict contributions from sources located outside the urban area (bck). Panel (b) displays the total average BC concentrations, accounting for contributions from all these sources. Black rectangles in the plots represent buildings. Please note the different scales used for panel (a) and panel (b).

among open fireplaces (10%), conventional stoves (8%), high-efficiency stoves (8%), and advanced stoves and boilers (9%). Since there was no available data for a more detailed distribution of these emissions, we applied a consistent approach to

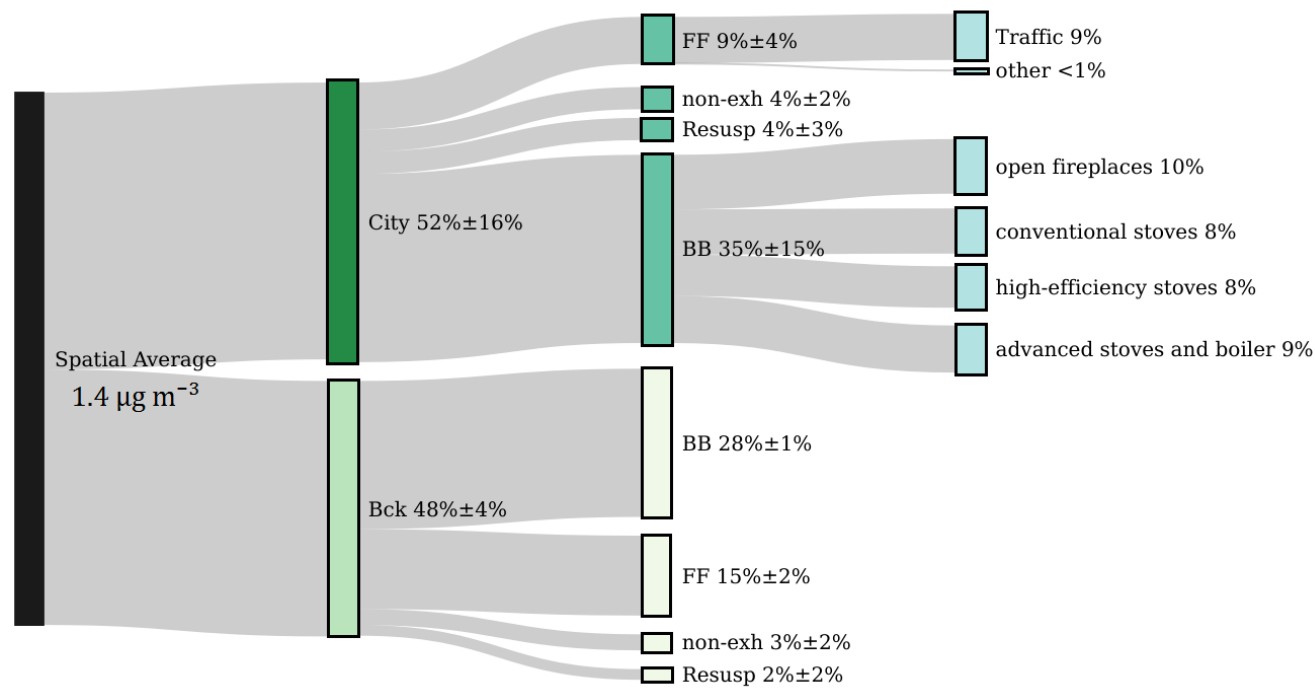

**Figure 9.** Sankey plot showing the percentage distribution of BC concentrations by source area and emission category.

spatially allocate all BB sources. As a result, the simulated BC concentrations align with the same percentage breakdown as the emission inventory. These results indicate that the urban contribution accounts for 52% ± 16% of the total average BC concentration, with the related uncertainty estimated using the root sum of squares method, assuming that the uncertainties associated with each quantity are independent of each other, that the relationship between the quantities being summed is linear, and that the uncertainties follow a Gaussian distribution. Regarding long-range transport, it contributes 28% ± 1% to BB emissions, 15% ± 2% to FF emissions, 3% ± 2% to non-exhaust emissions, and 2% ± 2% to resuspension emissions, with uncertainties estimated similarly to the urban contributions. A summary of these percentages is depicted in the Sankey diagram presented in Figure 9. Compared to the urban simulations, the uncertainty associated with long-range transport is lower, especially for BB. This reduction is due to the smaller discrepancies observed in the different simulated scenarios compared to the city scale results. Specifically, when considering BB at a regional scale, changes in the spatial distribution of emissions yielded almost identical results when assessing concentrations in Modena. This suggests that when using a regional chemical transport model to simulate long-range transport to an urban area at a resolution of 3 km, the spatial distribution of emissions in the surrounding region becomes a less critical factor. Furthermore, the effect of minimum/maximum BC speciation factors seems to have a secondary effect on long-range transport.

### 4.2.4 Contribution of traffic sources

Building on the general source apportionment previously reported, the availability of detailed traffic information facilitated a more detailed investigation of the traffic contribution. As outlined in section 2.5.1, five different sources were used to estimate traffic emissions. These sources include modelled traffic flows on a dense road network (Figure 1), historical traffic counts from induction loop spires at key intersections, a comprehensive breakdown of the fleet composition, an assessment of annual kilometres travelled by each vehicle class, and speed-dependent EF that take into account road and vehicle characteristics. Three types of traffic emissions were identified: exhaust, non-exhaust, and resuspension. Each type was simulated as a separate source in the GRAL model, allowing for a direct assessment of their individual contributions within the city. Additionally, dispersion simulations were conducted separately for the two representative vehicle categories (light and commercial vehicles) simulated by the traffic model. The results indicate that when the traffic contributions are aggregated, 50% can be attributed to exhaust emissions, with light duty vehicles accounting for the majority (38%). The remaining 50% is equally divided between non-exhaust and resuspension emissions, each contributing 25% (Figure 10).

Few studies in the literature have explicitly analysed the different contributions of exhaust, non-exhaust and resuspension emissions for BC or EC, making a comprehensive comparison difficult. Among the studies that explicitly refer to BC, Lugon et al. (2021) investigated the dispersion of traffic emissions in a limited area of Paris using comparable emission tools and modelling techniques to those used in this paper, showing a very similar breakdown between the three components (approximately 50% to exhaust emissions and the remaining 50% to the sum of non-exhaust and resuspension emissions). Other researches have combined source apportionment methods with chemical analysis to distinguish the contributions of different sources. For example, Demir et al. (2022) investigated exhaust and non-exhaust traffic emissions in a road tunnel using EC fractions and the positive matrix factorisation method. Their results show that non-exhaust emissions contributed up to 37% of the total carbonaceous species emitted by vehicles. In contrast, Karakaş et al. (2023) assumed that EC is only generated by fuel combustion or resuspended by traffic circulation, excluding contributions from tire, brake, or surface wear. Therefore, the methodology and assumptions used to determine BC in non-exhaust emissions can significantly impact the final results, contributing to greater uncertainties.

A closer comparison of the present work with that of Lugon et al. (2021) reveals that the similarities between these two studies can be attributed to the EFs from the EMEP guidebook, which both studies used as a reference. However, it is important to critically note that these factors are primarily based on a literature review of publications from 1995 to 2002, which were largely based on indirect estimates of total suspended particulates (TSP) and $PM_{10}$. Recent reviews of direct measurements of tyre EFs have reported values approximately 4.5 times lower than the EMEP guidebook value for $PM_{10}$ and 42 times lower for $PM_{2.5}$ (Charbouillot et al., 2023; Harrison et al., 2021). Conversely, experimental laboratory tests conducted in the latest years (Kim et al., 2022; Lyu and Olofsson, 2020) confirm the reference BC EFs for brake wear reported in the EMEP guidebook, showing similar or even higher values, with discrepancies of around 15%.

For road wear EFs, the methods available in the literature are far from exhaustive and their estimates are of the lowest quality compared to tyre and brake wear emissions (Ntziachristos and Boulter, 2019). Estimates of this component have been provided

by various authors (Woo et al., 2022; Amato, 2018; Gehrig et al., 2010; Thorpe and Harrison, 2008), but none of them explicitly refer to the BC or EC, making comparison with the EMEP guidelines difficult. Nevertheless, it should be noted that the contribution of road wear, compared to the other sources, is generally modest.

To explore the uncertainty associated with the attribution of the non-exhaust estimates reported in this study, an additional scenario was tested for each of the three components (tyre, brake and road wear). For tyre wear emissions, a correction factor of 4.5 was applied to the TSP EFs, as suggested by Charbouillot et al. (2023), to align with recent direct measurements, resulting in an average BC EF of $0.38 \, \mathrm{mg \, km^{-1} \, veh^{-1}}$. For brake emissions, the EFs were increased by 15%, resulting in an average BC EF of $0.25 \, \mathrm{mg \, km^{-1} \, veh^{-1}}$. As it was difficult to compare road wear emissions with recent studies, the associated
uncertainty was conservatively estimated by applying a correction factor of 4.5, similar to that used for tyre emissions. This resulted in BC EF values ranging from 0.35 to $0.02 \, \mathrm{mg \, km^{-1} \, veh^{-1}}$. The results show that when these correction factors are applied, the contribution of tire wear emissions decreased from 20% to 5%, while brake wear concentrations increased by 1%. For road surface emissions, the contribution varies from 5% to almost negligible when emissions are increased or reduced by 4.5, respectively.

The uncertainties reported in Figure 10 for tire, brake and road wear are computed as the standard deviation of the reported scenarios, resulting in 11% for tire, 1% for brake, and 2% for road wear emissions. The overall uncertainty of the non-exhaust concentrations is then calculated using the root sum of squares method, as previously described. Similarly, the uncertainty attributed to the BC resuspension is represented by the standard deviation between the scenarios simulated following the methodology proposed by Amato et al. (2012) and those performed with the NORTRIP methodology (see section 2.5.1).

In order to further explain the percentage breakdown of the subcategories to the exhaust emissions, the methodology used to calculate different contributions is resumed. The representative EF for the two simulated vehicle classes (light and commercial vehicles) is calculated by averaging the EFs of all subcategories, weighted by a combination of the fleet composition and estimated annual mileage per vehicle class. This integrated approach allowed the composition of the registered fleet to be adjusted to reflect an actual average presence of vehicles on the road (see Figure S14 for the fleet composition resulting from this calcu-
lation). The same approach is then used to calculate the contribution of each subcategory (vehicle type, fuel and Euro standard) to the total emissions of either light or heavy duty vehicles. From this analysis, passenger cars (PC) are the main contributor to the exhaust component accounting for 35% of total transport emissions, while mopeds and motorcycles, light commercial vehicles, buses and trucks contribute 3%, 10% and 2% respectively. In addition, Euro 4 diesel passenger cars, following the aforementioned methodology, are responsible for the largest share of BC emissions in the urban environment, accounting for
19% of total transport emissions. Although their EFs are lower than those of vehicles with older emission standards (Euro 3, Euro 2, etc., see also Figure S15), their share in the actual fleet composition (number of registered vehicles adjusted by annual mileage per vehicle class) makes this vehicle class to account for a large contribution.

The uncertainty of the total exhaust contribution is estimated to be 16% based on the standard deviation of the three simulated scenarios considering different BC speciation factors (see Section 2.5.1). On the other hand, Kouridis et al. (2010), suggested
that the uncertainty of the exhaust EFs at a speed of $50 \, \mathrm{km \, h^{-1}}$ estimated following the Tier 3 approach, is around 0.001 $\mathrm{g \, km^{-1}}$ for both petrol and diesel passenger cars (corresponding to 53% and 7% of the weighted EFs used for the same cate-

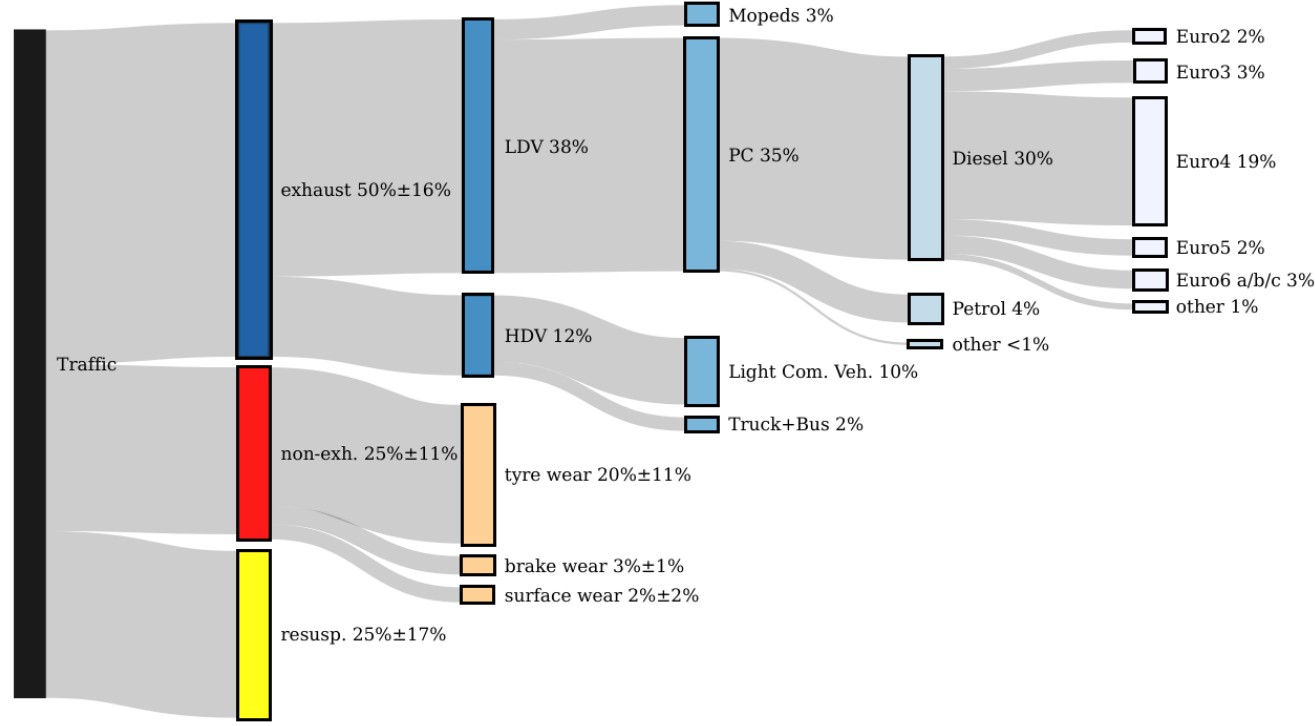

**Figure 10.** Sankey diagram illustrating the percentage distribution of BC emissions from urban traffic sources.

gories in this study), around $0.06\,\mathrm{g\,km^{-1}}$ (8%) for commercial vehicles and about $0.076\,\mathrm{g\,km^{-1}}$ (48%) for buses. No data is available for mopeds and motorcycles.

## 5 Conclusions

The present article discusses the outcomes of an integrated modelling and measurement approach designed to assess black carbon (BC) levels and sources in Modena, a city located in the Po valley and chosen as representative of a typical medium-sized urban setting of the region. The eBC measurements were performed with two multi-wavelength micro-aethalometers of the Aethlabs MA200 series during two winter periods: the first from 4 February to 7 March 2020, and the second from 26 December 2020 to 21 January 2021, at two different urban sites. One site was selected to represent typical urban background

conditions, while the other was specifically chosen to capture high traffic intensity. In parallel to the measurement campaign, we implemented a hybrid modelling approach integrating the NINFA modelling suite (built upon the chemical transport model CHIMERE) with the Lagrangian model GRAMM-GRAL. This approach aimed to provide spatially resolved insights into BC concentrations and to identify potential sources contributing to BC levels in the urban environment of Modena. GRAMM-GRAL was used to reconstruct BC concentration fields at high resolution (4 m), taking into account local sources and the

presence of obstacles (e.g. buildings, bridges or portals) in the flow field reconstruction. Conversely, NINFA was used to estimate the influence of external sources within the urban environment of Modena on BC concentrations.

Absorptions estimated with the MA200 instruments were further partitioned using the Multi-Wavelength Absorption Analyser (MWAA) model, allowing the distinction between different components, namely FF (emitted from fossil fuel sources), BB (from biomass burning) and BrC (brown carbon). The same partitioning was also used to differentiate eBC concentrations
between fossil fuels (eBC-FF) and biomass combustion (eBC-BB). The absorption related to BrC showed a consistent and remarkable low presence throughout the day in both periods (around $0.5 \ \mathrm{Mm}^{-1}$ in the first, $1.2 \ \mathrm{Mm}^{-1}$ in the second). On the other hand, FF, a marker for traffic-related emissions, showed distinct diurnal peaks during rush hours, in the morning (07:00-08:00 GMT) and in the evening (18:00-20:00 GMT) for both periods, with values up to $187.2 \ \mathrm{Mm}^{-1}$ (corresponding to 18.5 $\mathrm{\mu g \, m}^{-3}$ of eBC-FF concentration) at the urban traffic site. In contrast, BB showed less diurnal variability, with a minimum in
the early afternoon (15:00-17:00 GMT) and a peak in the evening (20:00-21:00 GMT), and generally lower absorption compared to FF. A comparison of the two seasons showed significant differences. The period from 26 December 2020 to 21 January 2021 had higher FF and BB concentrations, likely due to different meteorological conditions such as lower wind speeds and increased atmospheric stability, leading to frequent thermal inversions and a higher frequency of stable atmospheric conditions. In addition, lower temperatures likely led to more biomass burning for heating, which increased eBC concentrations.

In the second part of the study, the concentrations reproduced by the hybrid modelling system were compared with the observations recorded by the MA200 instruments. During the period from 4 February to 7 March 2020, the hybrid system showed a robust performance, with an MB of $0.02 \ \mathrm{\mu g \, m}^{-3}$ (+2%) at the urban background site and $-0.12 \ \mathrm{\mu g \, m}^{-3}$ (-5%) at the traffic site. The linear correlation coefficients were 0.51 and 0.62 respectively. Importantly, the daily concentration peaks were consistently captured even under varying meteorological conditions, including periods of high wind speed. However, during
the period from 26 December 2020 to 21 January 2021, a contrasting behaviour was observed between the traffic and background stations. At the urban background site, measurements were only available for the period from 26 December 2020 to 7 January 2021, which included 7 holidays out of 13 measurement days. Under these conditions, the model showed a tendency to overestimate concentrations from biomass burning emissions, especially during the early part of the day. This effect was similar to that observed in the first period, but more pronounced due to the increased influence of the holidays. For this station
the MB increased to $0.25 \ \mathrm{\mu g \, m}^{-3}$ (+15%) and the correlation coefficient decreased to 0.38. Conversely, measurements at the traffic station covered the entire period from 26 December 2020 to 21 January 2021. Throughout this period, the meteorological conditions were favourable for the accumulation of pollutants, characterised by the presence of high pressure systems and thermal inversions. In such circumstances, the model struggled to fully capture the complex meteorological dynamics, leading to a notable reduction in the mean bias ($-0.92 \ \mathrm{\mu g \, m}^{-3}$, -22%) and correlation coefficient (0.34). In addition, the presence of
occasional high emission sources in close proximity to the traffic monitoring site may have contributed to increased eBC concentrations. Despite the decrease in performance during the second period, the statistical metrics considered in this evaluation, including FAC2, NAD, FB and NMSE, met the established acceptance criteria for urban dispersion modelling. Furthermore, these results are consistent with metrics from similar studies simulating BC concentrations in urban areas.

The same system was used to identify the impact of different sources on the spatial average BC concentrations, both inside and outside the urban area of Modena. The analysis showed that the city itself contributes to 52% ± 16% of the total average BC concentration, while background sources account for about 48% ± 4% of the local concentrations. A detailed breakdown of the city's contribution showed that BB is responsible for 35% ± 15%, FF for 9% ± 4%, non-exhaust emissions for 4% ± 2% and resuspension for 4% ± 3%. In particular, FF is mainly attributed to traffic sources, with other FF sources playing a negligible role. On the other hand, BB emissions are evenly distributed between open fireplaces, conventional stoves, high efficiency stoves and advanced stoves, as detailed in the Arpae local emission inventory. As for long-distance transport, it contributes 28% ± 1% to BB emissions, 15% ± 2% to FF emissions, 3% ± 2% to non-exhaust emissions and 2% ± 2% to resuspension emissions. Finally, when analysing the traffic-related concentrations, we found that 50% can be attributed to exhaust emissions (in particular 19% of the total for Euro4, diesel passenger cars), with the remaining 50% divided equally between non-exhaust and resuspension sources, each contributing 25%. However, it is important to emphasise that significant uncertainties remain in the estimation of non-exhaust emissions, particularly for the fraction attributed to tyre wear. Different scenarios tested in this study show that the application of correction factors derived from recent literature can significantly change the estimates. Specifically, the contribution of tyre wear varies from 20% to 5%, brake wear varies from 3% to 4% and road surface emissions vary from 5% to almost negligible.

In future developments, this research can serve as a valuable basis for more comprehensive air quality management strategies in Modena and similar medium-sized urban areas in the Po valley. Ongoing monitoring and assessment of BC concentrations, together with a deeper understanding of the sources contributing to BC levels, can provide valuable insights for policy makers and urban planners seeking to reduce the impact of BC on air quality and public health in the region.

## Appendix A: Symbol and acronyms

**Table A1.** Description of symbols and acronyms used in the text.

| Symbol | Description |
|---|---|
| $\alpha$ | Absorption Ångström exponent |
| $b_{abs}$ | Aerosol absorption coefficient |
| $b_{abs,BC}^{BB}$ | $b_{abs}$ from BC by biomass burning |
| $b_{abs,BC}^{FF}$ | $b_{abs}$ from BC by fossil fuel combustion |
| $b_{abs}^{BrC}$ | $b_{abs}$ from brown carbon |
| BB | Biomass burning |
| BC | Black carbon |
| BrC | Brown carbon |
| CCN | Cloud condensation nuclei |
| $C_{ref}$ | multi-scattering correction factor |
| CTMs | Chemical Transport Models |
| EC | Elemental Carbon |
| EEA | European Environmental Agency |
| EF | Emission Factor |
| FF | Fossil fuel |
| IN | Ice nuclei |
| IR | Infrared |
| MAAP | Multi-angle absorption photometers |
| MAE | Mass absorption efficiency |
| MWAA | Multi-wavelength absorption analyser |
| NMVOCs | Non-methane volatile organic compounds |
| PBL | Planetary boundary layer |
| PC | Passenger cars |
| PM | Particulate matter |
| PTFE | Polytetrafluoroethylene |
| RB | Rural background |
| SNAP | Selected nomenclature for air pollution |
| UB | Urban background |
| UT | Urban traffic |
| UV | Ultraviolet |
| WHO | World health organization |

*Code and data availability.* The source code for CHIMERE, GRAMM-GRAL and the emission model VERT used in this study are available at the following permanent link: https://doi.org/10.5281/zenodo.13255628. Raw *in-situ* aerosol absorption data from MA200 devices for Modena during the study period is available at https://doi.org/10.5281/zenodo.8140250. A movie illustrating the diurnal variability of Fossil Fuel - BC, Biomass Burning - BC, and their combined sum for the two simulated periods is avaialble at https://doi.org/10.5281/zenodo.12960786.

*Author contributions.* GV designed the study in collaboration with AB, performed the CHIMERE and GRAMM-GRAL simulations, and wrote the paper. AB acquired the funds for the micro-athalomter measurements, performed the measurement campaigns and provided support in the data analysis. MS, ST, and GG provided essential resources and contributed to data interpretation. All co-authors review, commented and contributed to the manuscript.

*Competing interests.* The authors declare no competing interests.

*Acknowledgements.* This study was supported by the project 'Black Air' (CUP E94I19001080005) funded by the University of Modena and Reggio Emilia and the *Fondazione di Modena* under the programme "Fondo di Ateneo per la Ricerca 2019". This work has also been partially funded by the "Ecosystem for sustainable transition in Emilia Romagna (ECOSISTER)" project, identified with code ECS_00000033, funded by the European Union NextGenerationEU programme—Call for tender n. 3277 dated 30 December 2021, Award Number: 0001052 dated 23 June 2022—under the National Recovery and Resilience Plan (NRRP) Mission 4, Component 2, Investment Line 1.5: "Establishing and strengthening" of "innovation ecosystems for sustainability", building "territorial leaders of R&D". Carla Barbieri and Enrica Canossa from ARPAE are kindly acknowledged for hosting the micro-aethalomters in the air quality monitoring stations and for granting full access to these sites. We would like to extend our appreciation to Dario Massabò and Vera Bernardoni for the application of the MWAA model to micro-aethalomters measurements used in this study. The municipality of Modena is also kindly acknowledged for providing the vehicular traffic data.

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
