# Peer review of "Measurement report: Source attribution and estimation of black carbon levels in an urban hotspot of the central Po valley: An integrated approach combining high-resolution dispersion modelling and micro-aethalometers"

_EGUsphere, 2023_

## Author Response (AR1)

We sincerely thank both reviewers for their thorough evaluation of our manuscript and for their valuable comments and suggestions. Reviewer 1's insights highlighted several unclear sections and weaknesses, which we believe we have addressed in the revised manuscript. Similarly, reviewer 2's detailed feedback was instrumental in improving the clarity and robustness of our study. We believe that these revisions have improved the overall quality of our work, and we greatly appreciate the reviewers' contributions to this process.

In our response, we have used the following color scheme: the reviewer's comments are in red font, our responses are in black font, and excerpts from the revised manuscript are in blue font. Text that has been deleted is shown with a strikethrough. Please note that references to specific lines in the manuscript correspond to the author's track changes file.

REVIEW

Veratti et al.,

Source attribution and estimation of black carbon levels in an urban hotspot of the central Po Valley: An integrated approach combining high-resolution dispersion modelling and microaethalometers

ACP

STAGE 2

General comments

Veratti et al. describe an interesting combination of in-situ monitoring and modelling application to a mid-sized city in the Po valley. The results clearly demonstrate the need to treat the contribution to black carbon (BC) of biomass burning (BB) and traffic (the sole source of fossil fuel – FF) on different spatial scales.

The manuscript is well written (even if somewhat long-ish) and deserves publication after the comments below are addressed. It may become a tool for many municipalities to analyze the sources of primary air pollution and measure the efficiency of abatement.

The especially important comments are the ones that need to be addressed at:

L 148-151

L 491-495

L 587-647

L 634-636

L 638-642

Supplement, Fig. S10

Specific comments

1)	Lines 20-26: The terminology on BC and EC can be shortened and Petzold et al. (2013) cited.
The text has been shortened and only the essential information has been retained. See lines 20-26.

We changed the text as follows:
~~The vocabulary associated with BC is extensive and many uncertainties arise from the definitions and measurement methods used for BC (Petzold et al., 2013; Bond et al., 2013; Grange et 1. According to the recommendations of Petzold et al. (2013), BC is the preferred term when formally characterising a material that is ideally light absorbing and composed predominantly of carbon . Conversely, EC is a pure form of carbon, consisting only of carbon atoms not bonded, and this term should be used instead of BC when reporting data derived from methods that specifically measure the carbon content of carbonaceous matter~~ (Petzold et al., 2013).

2)	L 38-51: When discussing the climate effects of BC, cite the latest IPCC report.
The citation to the latest IPCC report has been added, Line 44.

3)	L 75, "This approach…": This sentence needs to be replaced by a longer explanation of how filter absorption photometers work. There is a difference between the true absorption measurement (such as photoacoustics or photothermal interferometry) and proxy measurements with filter photometers. Parametrization of corrections needs to be addressed only in the context of the multiple scattering parameter – value 1.3 is used later on.
A more comprehensive explanation of the functioning of absorption photometers has been incorporated into the introduction of the main text. See lines 79-93.

We changed the text as follows:
 In this approach, sample air is passed through a filter tape where aerosol particles are collected. Optical filter photometers measure light transmission, or a combination of reflection and transmission, through the sample-loaded filter and calculate the attenuation coefficient from the rate of change of attenuation over time. However, the attenuation coefficient can differ significantly from the true aerosol absorption coefficient due to two main artefacts. The first is the enhancement of the optical path, and hence the enhancement of light absorption of the deposited particles, due to the multiple scattering of the light beam at the filter fibres and between particles and fibres. The second is the loss of instrument sensitivity with increasing particle loading. To overcome these limitations, several empirical corrections have been proposed in the literature. For instance, the Cref factor is used to correct for the multiple scattering effect, while the f(ATN) function is applied to compensate the loading effect. Typically, the Cref factor is assumed a priori, but it can also be determined experimentally through multiinstrument colocation. Conversely, the loading correction function f(ATN) is increasingly estimated online using dual-spot technology (Drinovec et al., 2015), or it can be estimated offline using dedicated algorithms (Weingartner et al., 2003; Virkkula et al., 2007; Park et al., 2010). The specific mass absorption efficiency (MAE; also known as the mass absorption cross section, MAC), can then be used to convert the aerosol light absorption coefficient  into the light-absorbing carbon mass concentration.

4)	L 85: Add the revision of the EU Air Quality Directive, as it requires BC measurements to be taken.
A reference to the revision of the EU Air Quality Directive has been added (lines 111-115).

We changed the text as follows:

In 2022, the European Commission proposed revisions to the Ambient Air Quality Directives to bring European Union air quality standards more closely in line with WHO recommendations (European Council, 2022). These revisions highlight the importance of monitoring emerging pollutants such as BC to support scientific understanding of their effects on health and the environment, in line with the WHO's guidance.

5)      L 109-111, "… or attempted to apportion the sector-specific contribution": This is unclear. Apportionment means ascribing a measured parameter to sources. I think the authors mean: the contribution of sources in the model to ambient concentrations at a site. Please reword.

The sentence has been rephrased (lines 140-141).

We changed the text as follows:

However, none of these studies focused on assessing total BC concentrations over an entire urban area at high resolution, or attempted to  identify the contribution of different sources at monitoring sites.

6)      L 131 and elsewhere: The temperature inversions are mentioned – please add plots demonstrating this in the Supplement.

In the supplementary material, vertical profiles of soundings conducted at S. Pietro Capofiume, located 50 km from the Modena urban area (see Figure 1 of the main text), are presented for January 3rd, 4th, 13th, 14th, 18th, and 19th, 2021, at 00:00 and 12:00 GMT, spanning from Figure S8 to S10.

7)      L 142: The inlet temperature of 30 C may lead to losses in BrC and/or coatings from aerosol. There needs to be some explanation on the use of such a high temperature for winter measurements.

The decision to maintain an inlet temperature of 30°C for winter measurements was based on practical considerations. Due to space limitations in the two cabinets, we used existing glass manifold inlets originally designed for reactive gas monitors, which naturally maintain the inlet air at 30°C. This configuration was applied consistently to the MA200s throughout the sampling period to mitigate the effects of humidity variations during aerosol sampling. Previous studies have shown that rapid changes in relative humidity can significantly alter the absorption coefficient of filter-based instruments (Düsing et al., 2019), so our aim was to minimize these effects. As an additional benefit, this approach could reduce to some extent the uncertainties associated with the comparison of simulated and measured BC concentrations (see comment number 12 and related answer).

8)      L 148-151: The "additional" correction factor of 1.3 needs a thorough explanation – see comment above (L 75). The MAC values, as reported in the Supplement are quite high when compared to the ones in the literature and the ACTRIS recommended ones (~10 m2/g at 635 nm). An explanation on the choice of the multiple scattering parameter 1.3 and the MAC values, and at least a comparison with the published ones (Zanatta et al., 2016; Savadkoohi et al., 2023) is required.

We have further elaborated on the explanation of the multi-scatter correction factor, expanding on the details provided in the introduction (lines 79-93). We have also provided additional clarification on the rationale for selecting a Cref of 1.3 (lines 183-199). In addition, we have included supporting information on the use of default MAE values in our study (lines 200-205).

To address concerns regarding MAE values, we have performed a comprehensive comparison with the published literature, using data from sources such as Zanatta et al. (2016) and Savadkoohi et al. (2023; 2024). This comparative analysis is detailed in Table S1, where the MAC values used in our study are compared with

those reported in the existing literature. It is noteworthy that, although the MAE values for Modena tend to be higher than those reported elsewhere, they fall within the range observed by Gilardoni et al. (2020) and Mousavi et al. (2019) in studies conducted in the Po Valley. Furthermore, our MAE value at 880 nm is consistent with values reported for other European urban areas such as Athens (Savadkoohi et al., 2024), Zurich and Bern (Grange et al., 2020), as well as urban traffic and background locations in Leipzig and Prague for MAE values at 637 nm (Savadkoohi et al., 2024).

We changed the text as follows:
For Lines 79-83 see comment 3).

Lines 183-199:
As a result, we opted for a constant Cref of 1.3 was chosen to account for multiple light scattering within the filter fibres and between deposited particles and the filter, as suggested by the manufacturer (Aethlabs, 2024). To support this decision to mimic the response of the AE33 aethalometer (Aethlabs, 2024). In addition, to convert the aerosol light absorption coefficient to equivalent mass concentration, we relied on the wavelength-specific MAE values provided in the MA200 reference manual and reported in Table S1. In support of these decisions, a recent instrument intercomparison performed at an urban background site in Athens (Stavroulas et al., 2022) showed limited differences between the in terms of eBC concentrations between a MAAP and the two MA200 instrument units used in this study, set with Cref of 1.3 and a MAAP default MAE values (linear slope of 1.00 in winter and 1.07 in summer, with an $r^2$ of 0.92 for both the seasons). Furthermore, as site-specific MAE values for Modena were not available, we relied on the wavelength-specific values provided in the reference manual and reported in From the same intercomparison campaign, two MA200 units were also compared with an AE33 aethalometer, showing strong agreement during winter (linear slope between 0.91 and 0.97 and $r^2$ of 0.97 for both devices). Furthermore, recent studies corroborate the results of the Athens intercomparison showing consistent findings when comparing the AE33 and the MA200. For example, Blanco-Donado et al. (2022) observed a linear regression slope of 0.97 and a $r^2$ of 0.93 during a 3-day intercomparison campaign in a suburban area of Barranquilla (Colombia). Similarly, Khan et al. (2024) reported a linear regression slope of 0.986 and a $r^2$ of 0.97 for a 14-hour intercomparison conducted in an urban background of Vilnius (Lithuania).

Lines 200-205:
Table S1 in the supplement S1 in the supplementary material presents a comparison of the MAE values used in this study with those measured across various European locations. Although the values used for Modena are generally higher than those reported in the literature, they are consistent with previous observations conducted in the Po valley, such as Gilardoni et al. (2020) and Mousavi et al. (2019). Furthermore, the MAE value at 880 nm used for this study is in the range of values reported for other European urban areas, such as Athens (Savadkoohi et al., 2024) , Zurich and Bern (Grange et al., 2020), or in the range observed for urban traffic and background sites in Leipzig and Prague for MAE values at 637 nm (Savadkoohi et al., 2024).

9)      L 152-158: Was the aggregation of the data achieved by recalculation or averaging? The Bigi et al. (2023a) paper is not clear on that. Additionally, Bigi et al (2023b) was in the meantime accepted in ACP.
The data aggregation process involved recalculating the original data on a 5-minute basis, after which they were averaged to derive hourly values. The reference to Bigi et al., (2023) has been updated with the version of the manuscript accepted in ACP.

10)      L 177, "… by multi-wavelength fitting of 1 …": "… by multi-wavelength fitting of Eq. 1 …"?
Yes, there was a typo in the original version of the manuscript. The text has been amended (line 241).

11)      L 187, Table 1: Is the "sensor height" meant above ground? If so, please mention.

Yes, 'sensor height' refers to the height of the sensor above the ground. The text has been revised accordingly. (Table 1).

12)      L 201-202 and elsewhere: BC was treated as inert in GRAMM-GRAL and NINFA. While this is true for BC, it is not true for the coatings that might accumulate on the BC particles. These coatings increase the MAC and the response of filter photometers overestimated the BC mass concentration (Kalbermatter et al., 2022). The Po valley is a location where BC is coated fast. Modelling the coatings is extremely difficult, but the authors should estimate the uncertainty induced by potential coatings.

We acknowledge that neglecting the accumulation of coating material on BC particles could lead to an overestimation of BC mass concentration, especially for emissions originating outside Modena for which aging processes likely occur. This is a limitation of our current study, as our hybrid modeling system treated BC as inert. As the reviewer pointed out, modeling the mixing state, layer thickness, and composition remains a major challenge (Curci et al., 2019).

Global studies report a range of BC absorption efficiency enhancement (Eabs) from almost negligible to 2.4, with the most relevant range for urban environments falling between 1.2 and 1.6 (Moffet et al., 2009; Bond et al., 2006; Liu et al., 2017; Schwarz et al., 2008). While our analysis indicates that roughly half of the modeled BC concentrations originate from Modena itself, where we assume minimal coating effects (especially for the traffic location), it's plausible that BC from surrounding areas undergoes aging, potentially increasing absorption. To account for this uncertainty, we investigated the effect of varying the MAE within a range of ±20%, reflecting half of the documented Eabs enhancement range. The results of this sensitivity analysis are presented between lines 801 and 828 in the main text and in Table S3 of the supplementary material.

We changed the text as follows:

Lines 801-828:

A limitation of our study is that the modelled BC concentrations are treated as inert and no external or internal mixing coatings were simulated by the hybrid modelling system. While the simulation of coating absorption is particularly challenging (Curci et al., 2019), especially for Lagrangian models (Doran et al., 2008; Holmes and Morawska, 2006), omitting this aspect could introduce potential uncertainties during model validation when assuming reference MAE values for aethalometer measurements. In fact, different mixing states can lead to an enhancement of the BC absorption efficiency (Eabs) compared to the theoretical pure form of BC. Studies carried out at different locations around the world have reported a wide range of Eabs values, from almost negligible (Cappa et al., 2012) to 2.4 (Peng et al., 2016), with most falling in the range of 1.2-1.6 (Moffet and Prather, 2009; Bond et al., 2006; Liu et al., 2017; Schwarz et al., 2008). To account for the uncertainty associated with MAE values when comparing modelled and measured concentrations, we investigated the impact of varying MAE. In particular, although we found that about half of the modelled BC concentrations in Modena originate from the city itself (as detailed in section 4.2.3), where we assume a negligible coating, especially when analysing concentrations close to emission sources, it is reasonable to expect that regional BC emissions originating from outside Modena may undergo ageing processes and thus experience absorption enhancement. Taking these factors into account, we examined the effects of adjusting the MAE within a range of ± 20%, which represents half of the average absorption enhancement range (1.2-1.6) documented in numerous studies in the literature. Figure S13 in the supplementary material shows the linear regression between modelled and measured BC concentrations, using data from both simulated periods and the reference MAE values presented in Table S1. Setting the intercept to zero yields slope values of 0.83 and 0.65 when evaluating total BC concentrations at background and traffic sites. At the same sites, the RMSE of the regression model is 1.0 and 1.7, respectively. Focusing only on the contribution of biomass burning, the slope of the regression line is 0.98 and 0.77 at background and traffic sites, respectively, with corresponding RMSE

values of 0.7 and 1.0. For the fossil fuel contribution, the slope of the regression line is 0.49 and 0.56 and the RMSE is 0.63 and 1.2 at background and traffic sites, respectively. By increasing the MAE by 20% compared to the values reported in Table S1, both the slope of the regression line and the RMSE of the linear model increase by about 17%. Concurrently, the NMB between modelled and observed concentrations shifts from +7% to -11% at the urban background site and from -0.16% to -30% at the traffic site when assessing total BC concentrations. Conversely, a 20% reduction in the MAE results in a reduction of approximately 25% in both the slope of the regression line and the RMSE of the linear model. This variation in MAE results in an NMB of +34% at the urban background site and +5% at the traffic site when analysing total BC concentrations. The outcomes of the linear regression, together with the NMB values for the three scenarios analysed, are presented in Table S3 in the supplementary material. These results include assessments of total BC concentrations as well as contributions from biomass burning and fossil fuels.

13)     L 222, "… 10 nm to 40 m.": I suspect that the authors did not mean spectacularly giant aerosols – µm?
Yes, there was a typo in the measurement unit. The text has been revised. Line 279.

14)     L 287, "City emissions" subsection: The description of how EFs were obtained is fairly detailed which is to be commended. The Supplement lists EF diurnal profiles in Fig. S10, but without the unit. It would be wise to report the EF values somewhere, especially in light of comments below.
For traffic emissions, emission factors (EFs) depend on vehicle speed, which varies from street to street. This makes it difficult to summarize a representative emission factor for the whole city. However, to provide more insight into the EFs for different fuel types and Euro emission standards, Figure S15 in the supplementary material shows the exhaust EFs used in this study as a function of driving speed.
For traffic non-exhaust emissions, representative EFs are given in the newly added section 4.2.4 (lines 930, 931 and 933). The EFs for traffic re-suspension, according to the methodology used, are given in Table 2 and section 2.5.1 of the main text. Emissions from other sectors were derived from the INEMAR (2023) local emission inventory and therefore derived from annual total expressed as tons per year, but for completeness, representative emission factors and their associated 95% confidence intervals used to derive total emissions reported in the inventory, are given in Table S2 in the supplementary material.

15)     L 438, "Meteorology" subsection: This section is very detailed and lacks the parameter which is mentioned as being important later on: the PBL height. Consider moving some of it in the Supplement and adding PBL comparison here.
We agree with the reviewer that the PBL height is a crucial meteorological parameter influencing dispersion processes, making its assessment essential for local-scale modeling. Unfortunately, direct measurements of the PBL height in Modena are not available, making a direct comparison with modeled values impossible. However, we recognize the importance of this parameter and have taken steps to address this limitation within our study.
To provide useful information on the ability of the employed modeling system to reproduce the PBL height, we compared the simulated PBL height, as predicted by NINFA, with estimates derived using the bulk Richardson number from sounding data at 00:00 and 12:00 GMT in the nearby area of S. Pietro Capofiume (Fig. 1). These comparisons are detailed in the revised manuscript (lines 594-625), ensuring that the assessment of this critical parameter is adequately represented.
Additionally, to offer insights into whether GRAL can realistically reproduce the PBL height, we have included a qualitative comparison of the PBL height simulated by GRAL. This analysis indicates that on most days, GRAL produces values close to the observations conducted in the rural area of S. Pietro Capofiume, suggesting that the results may be realistic. However, there are limited episodes during nighttime (00:00 GMT) from 15

February to 7 March 2020 and during daytime (12:00 GMT) from 26 December 2020 to 21 January 2021 where the PBL height is likely overestimated.

Since the comparison between the simulated PBL height by GRAL and observations is qualitatively discussed, we have opted to keep the related plots in the supplementary material (Fig. S5 and S6). The discussion has been added to the main text at lines 594-625.

We changed the text as follows:

Lines 594-625:

Further analysis was carried out on the height of the planetary boundary layer (PBL), a key driver of atmospheric dispersion. Although direct measurements of the PBL height were not available for Modena, we made both quantitative and qualitative comparisons with observations taken in rural areas at the S. Pietro Capofiume (SPC) station, approximately 50 km east of Modena (see Figure 1 and Table 1). Specifically, we estimated the PBL height using the Richardson number (Ri) derived from sounding data at 00:00 and 12:00 GMT, using 0.25 as the critical value for identifying turbulent conditions (Lyons et al., 1964; Galperin et al., 2007; Grachev et al., 2013). These estimates were then compared with the PBL height simulated by NINFA, which also uses the Richardson number in its calculations (Troen and Mahrt, 1986). Figures S5 and S6 in the supplementary material provide an overview of this comparison for two periods: 15 February to 7 March 2020 and 26 December 2020 to 21 January 2021. For the first period, data was only available at 00:00 GMT, while for the second, data was also available at 12:00 GMT. The results show that between 15 February to 7 March 2020, NINFA generally underestimates the sounding estimates, with a MB of -100 m (-52%), resulting in a shallower mixing layer compared to the measurements. However, during the second period, the PBL height was better reproduced, with a limited MB of -47 m (-27%) at 00:00 GMT and -38 m (-12%) at 12:00 GMT, showing a robust performance in simulating the vertical structure of the atmosphere. Despite the more pronounced underestimation of the first period, NINFA showed similar performance to other meteorological models applied in the Po valley and other locations in Italy in reproducing the PBL height derived from soundings (Ferrero et al., 2011; Avolio et al., 2017).

For a qualitative comparison, the PBL height simulated by GRAL over the urban area of Modena is also included in Figures S5 and S6. Although a quantitative analysis between GRAL and soundings is not possible due to the fact that the measured area is outside the GRAL domain and various factors may cause differences in the vertical turbulence profile between urban and rural areas (e.g. urban heat island effects and anthropogenic heat sources), this comparison serves as a basis for hypothesising whether GRAL can realistically reproduce the PBL height during sounding time.

The results of the GRAL simulations show that during the first period the PBL height is in agreement with the sounding data on most days (16 out of 22). However, on 6 of the 22 days, GRAL significantly overestimates the measurements at 00:00 GMT, with values up to 800 m, which corresponds to the domain top internally set in the model code. During the second period, the PBL height modelled by GRAL generally matches the soundings data at both 00:00 GMT and 12:00 GMT, except for nine days when there is a significant overestimation (up to four times the observations) at 12:00 GMT. A more detailed analysis of these episodes shows that during the night (00:00 GMT), when stable or very stable atmospheric conditions are imposed on GRAL, the PBL height tends to match the values observed at SPC. Conversely, under neutral conditions, the simulated PBL height by GRAL overestimates the observations by a factor of three to six. This overestimation under neutral conditions is also evident at 12:00 GMT during the second period, with 6 out of 9 episodes occurring under these conditions.

Despite these challenges faced by GRAL, a common limitation for urban air quality models (Sokhi et al., 2022), the simulated PBL height appears realistic on most days. In addition, it is important to note that the limited number of observational data points prevents further conclusions being drawn for other times of the day.

16)    L 491-495, Fig. 4, Fig. 5, L577: The explanation about the thermal inversions and nearby sources causing high BC concentrations at the traffic site sounds overly simplistic. There is an obvious spike in 1-hour averages (!) also at the background station on 2 Jan 2021 and other periods of high concentrations appear at both sites as well. This seems like a meteorological effect, possibly non entirely linear. It may be that the models simply do not capture well the extreme events, this is known to happen with (more or less) linear models (see also comment below on the EFs). The discussion needs to be expanded, taking advantage of suggestions below.

We have critically analyzed the results, focusing more on the meteorological aspects that may have contributed to the high concentration peaks in Modena, and have extended the discussion accordingly.

In the revised version of Section 4.2.1, we have emphasized the challenges faced by the GRAL model in accurately simulating the PBL height over urban areas. While GRAL generally agrees with observations at 00:00 GMT and 12:00 GMT, it struggles to reproduce realistic PBL heights during certain sporadic episodes, particularly under neutral conditions at night. This limitation could hinder the model's ability to capture the very stagnant meteorological conditions typical of the Po Valley.

We acknowledge that different concentration spikes are present at both the stations and this is likely to result from complex meteorological phenomena that can occur in the study area. These meteorological situations are difficult for linear models such as GRAL to capture accurately, especially during extreme events. Based on your suggestions, we have expanded the meteorological discussion in our manuscript. In particular, we have detailed how factors such as strong inversion layers, temporary reductions in wind speed and complex urban meteorology contribute to elevated BC concentrations. The revised text highlights these factors more comprehensively and addresses both observed and modeled discrepancies.

Despite the meteorological explanations, certain concentration peaks, such as the one observed on 4 January, were recorded only at the traffic site and were dominated by fossil fuel contributions. This suggests the presence of local sources, such as high-emitting vehicles passing nearby or idling in nearby car parks. While we acknowledge the limitations of the GRAL model, we also believe that local sources may play a significant role in these specific cases.

Changes in the main text have been reported at lines 653-693:

[revised manuscript text omitted]

17)     The timeseries in Figs. 4 and 5 should include eBC separated into BC_ff and BC_bb. Makle the figures larger in y-direction for transparency.

Figure 4 and 5 have been modified following the reviewer's suggestion. The figures now display eBC and simulated BC divided by FF, BB and the sum of the two. In addition, the y-axes have been adjusted to better reflect the station concentrations, enhancing clarity and transparency.

18)     In addition, the regressions between BC, BC_ff, BC_bb should be shown. They have been performed as results are discussed in the text later on.

The linear regression analyses comparing total modeled and measured BC concentrations, along with those specifically for Biomass Burning-derived BC and Fossil fuel-derived BC, have been integrated into the main text between lines 814 and 828. These analyses are also visually represented in Figure S13 and further detailed in Table S3 for reference.

We changed the text as follows:
Lines 814-828:
Figure S13 in the supplementary material shows the linear regression between modelled and measured BC concentrations, using data from both simulated periods and the reference MAE values presented in Table S1. Setting the intercept to zero yields slope values of 0.83 and 0.65 when evaluating total BC concentrations at background and traffic sites. At the same sites, the RMSE of the regression model is 1.0 and 1.7, respectively. Focusing only on the contribution of biomass burning, the slope of the regression line is 0.98 and 0.77 at background and traffic sites, respectively, with corresponding RMSE values of 0.7 and 1.0. For the fossil fuel contribution, the slope of the regression line is 0.49 and 0.56 and the RMSE is 0.63 and 1.2 at background and traffic sites, respectively. By increasing the MAE by 20% compared to the values reported in Table S1, both the slope of the regression line and the RMSE of the linear model increase by about 17%. Concurrently, the NMB between modelled and observed concentrations shifts from +7% to -11% at the urban background site and from -0.16% to -30% at the traffic site when assessing total BC concentrations. Conversely, a 20% reduction in the

MAE results in a reduction of approximately 25% in both the slope of the regression line and the RMSE of the linear model. This variation in MAE results in an NMB of +34% at the urban background site and +5% at the traffic site when analysing total BC concentrations. The outcomes of the linear regression, together with the NMB values for the three scenarios analysed, are presented in Table S3 in the supplementary material. These results include assessments of total BC concentrations as well as contributions from biomass burning and fossil fuels.

19)      L 525: What is the "inherent underestimation of the traffic flows". Please elaborate. I understand that traffic counts are available.

Simulated traffic flows from the PTV model were combined with historical traffic measurement data from 400 induction loops for traffic light control on the urban and suburban road network, as well as radar Doppler data collected in winter 2016 near the urban traffic station, to estimate typical traffic patterns for both holiday and working days. In response to the reviewer's comment, we compared the traffic data estimated from this combination of simulation and measurement with actual traffic flows measured by 45 induction loop sensors located at traffic lights near the two monitoring stations. The comparison results confirmed the hypothesis of traffic underestimation between 18:00 and 21:00 GMT on 36 out of 57 simulation days.

The text has been modified between lines 726 and 730:

 In order to test this hypothesis in depth, we compared the measured traffic flows from 45 induction loop sensors located at traffic lights in the vicinity of the two monitoring stations with the traffic flows simulated by PTV VISUM on the reference roads around these stations. The results showed that on 36 out of the 57 simulation days, the traffic flows between 18:00 and 21:00 GMT were underestimated, with NMB of up to -18% for roads near the urban traffic station and up to -13% for roads surrounding the urban background site. Despite this underestimation, the overall agreement of model results with observations further confirms the effectiveness of a bottom-up approach in accurately representing traffic emissions.

20)      L 548-549, Fig 6: The reason for the 1-hour delay in the model relative to measurements needs to be investigated in more depth.

The one hour delay reported in Figure 7 mainly affects holidays rather than working days. We believe that the main factor contributing to this overestimation is the emissions modulation profile used for holidays (Figure S12), which appears to be inaccurate for the Christmas period. The traffic emission profile for these simulations, as previously mentioned, is derived from historical traffic measurement data from induction loops and radar Doppler data collected in winter 2016. These data are representative of average traffic situations rather than specific events like Christmas, and may not capture the unique and variable traffic patterns typical of that time of year. Additionally, the modulation profile for BB seems also to produce delayed BC concentrations with respect to observations during the holiday period, suggesting it may not accurately represent the behavior of domestic activities during Christmas. Conversely, we did not observe the same one-hour delay during working days, making it more difficult to attribute this concentration delay to meteorological conditions.

A brief explanation has been added to the main text at lines 751-752:

However, the magnitude of the night peak is well captured, albeit with a one hour delay, probably due to the emission modulation profiles imposed on GRAL during the Christmas holidays for both FF and BB (Figure S12), which do not seem to fully represent the observed pattern.

21)      L 551-554: The "divergent results" should be investigated and the analysis repeated for the time period shared between both stations. Do the differences remain?

As suggested by the reviewer, we have repeated the analysis for the time period common to both stations. The results indicate that at the traffic site, the model's performance is similar to that at the urban background site. Specifically, we observed a mean bias (MB) of 0.42 μg m⁻³, corresponding to +16% of the normalized mean bias (NMB). The Pearson correlation coefficient between modeled and observed values is 0.34, which closely aligns with the statistics reported for the urban background site. This behavior can mainly be attributed to three episodes where the model failed to accurately capture the daily trends. Between 27-28 December 2020 and on 6 January 2021, the model likely failed to reproduce local meteorological conditions, leading to high concentration estimates for most hours of these days. Conversely, the overestimation observed between 4 and 5 January 2021 was driven by the BB component at both stations. This may suggest that the concept of heating degree days, used to assign daily BB emissions, was not accurate for this specific episode.

The previous considerations have been added to the main text at lines 758-767:
When analyses are restricted to periods common to both sites, model concentrations at the traffic site show similar behaviour to the urban background site. The hybrid modelling system reports an MB of 0.42 μg m⁻³, corresponding to +16% of the NMB, while the Pearson's correlation coefficient between modelled and observed values is 0.34, which is very similar to the statistics reported at the background site (Table 7). This overestimation is mainly due to three different episodes occurring between 27-28 December 2020, 4-5 January 2021 and 6 January 2021. The first and last episodes are evident at both sites, with both FF and BB affected by an overestimation of the same magnitude in relative terms. This suggests that the local meteorology was not well captured by GRAL, probably resulting in a lower PBL height than the actual values. On the other hand, the episode between 4 and 5 January 2021 is dominated by an overestimation of the BB component only at both sites. In this case, the concept of heating degree days used to assign the daily BB emissions to each simulated day may not have accurately represented the actual emissions for that specific day.

22)      L 567: The reference to Figs. S1 and S2 is wrong. Please correct.
The original reference was intended to reflect the different temperatures measured in Modena during both simulated periods, rather than the daily modulation profile of BB emissions. Therefore, the incorrect references have been removed (Line 780).

23)      L 587-647, "4.2.3 Dispersion modelling based source apportionment": The Snakey plots are an important visualization tool. It is unclear what are the sources of the uncertainty (for example 52%+/-10% for city contribution to BC). Some of the uncertainties are very small. Please elaborate.
To enhance clarity, we have added details about the uncertainty in each simulated contribution (see lines 874-886). The reference value shown in the graph represents the average result across all simulated scenarios, while the corresponding uncertainty reflects the standard deviation, which captures the spread of these scenarios. We have also updated the total uncertainty for contributions at both urban and background scales. This was done using the root sum of squares method, which is commonly used to combine uncertainties in linear sums. Assuming that the uncertainties are independent, that the relationship between the quantities being summed is linear and that the uncertainties follow a Gaussian distribution, the uncertainty of the sum is calculated as the square root of the sum of the squares of the individual uncertainties.
The results of this analysis show that the standard deviation between simulations performed at the regional scale is generally lower than the same quantity at the city scale, especially for BB emissions. This indicates that when assessing the transport of emissions from regional sources to an urban area of interest at a horizontal resolution of 3 km, the spatial distribution of emissions in the surrounding area becomes a less critical factor. Furthermore, at the same scale, the influence of different BC speciation factors appears to have a secondary effect on long-range transport.

We changed the text as follows:

Lines 874-886

 These results indicate that the urban contribution accounts for 52% ± 16% of the total average BC concentration, with the related uncertainty estimated using the root sum of squares method, assuming that the uncertainties associated with each quantity are independent of each other, that the relationship between the quantities being summed is linear, and that the uncertainties follow a Gaussian distribution. Regarding long-range transport, it contributes 28% ± 1% to BB emissions, 15% ± 2% to FF emissions, 3% ± 2% to non-exhaust emissions, and 2% ± 2% to resuspension emissions, with uncertainties estimated similarly to the urban contributions. A summary of these percentages is  depicted in the Sankey diagram presented in Figure 9. Compared to the urban simulations, the uncertainty associated with long-range transport is lower, especially for BB. This reduction is due to the smaller discrepancies observed in the different simulated scenarios compared to the city scale results. Specifically, when considering BB at a regional scale, changes in the spatial distribution of emissions yielded almost identical results when assessing concentrations in Modena. This suggests that when using a regional chemical transport model to simulate long-range transport to an urban area at a resolution of 3 km, the spatial distribution of emissions in the surrounding region becomes a less critical factor. Furthermore, the effect of minimum/maximum BC speciation factors seems to have a secondary effect on long-range transport.

24)  L 595-596: Please add an online movie of the maps sowing modelled spatial distributions of diurnal profiles for BC_ff and BC_bb. This is super interesting.

A movie illustrating the diurnal variability of Fossil Fuel - BC, Biomass Burning - BC, and their combined sum for the two simulated periods has been included in the supplementary material. Additionally, the movie is available in the permanent repository at the following link: https://doi.org/10.5281/zenodo.12960786.

25)  L 634-636: I am very skeptical about the attribution of 50% of BC to non-exhaust emissions. This requires at least a paragraph of explanation, not a single sentence. This result is extreme and highly unexpected. At least street cleaning schedules need to be used to semi-quantitatively explain this with measurements.

In response to this concern, we have added a new section, 4.2.4 Contribution of Traffic Sources, to our manuscript (lines 888-941) to provide a more comprehensive explanation of the methodology used to estimate non-exhaust emissions. While our findings are consistent with those reported by Lugon et al. (2021), which used the same EMEP/EEA methodology, we acknowledge that attributing 50% of BC to non-exhaust emissions might seem extreme. Significant uncertainties are still associated with BC non-exhaust emission factors, particularly for tire emissions, which have a broad range of estimates in the literature.

To address these uncertainties, we tested additional scenarios for each component (tire, brake, and road wear) using correction factors derived from recent direct estimates of tire emissions (Charbouillot et al., 2023; Harrison et al., 2021), brake emissions (Kim et al., 2022; Lyu and Olofsson, 2020) and road surface emissions. The results showed that applying a correction factor derived from recent literature significantly alters the estimated contributions: tire wear emissions decreased from 20% to 5%, brake wear concentrations increased by 1%, and road surface emissions varied from 5% to almost negligible, depending on the correction factor applied. Although these additional scenarios do not provide definitive results, they help outline the range of uncertainties associated with non-exhaust emissions.

A semi-quantitative analysis of street cleaning material was not feasible, as this material was not collected during the sampling period. Furthermore, street cleaning in Modena is limited to certain pedestrian roads where outdoor markets occur, which are not representative of typical busy streets.

We changed the text as follows:

Lines 888-941:

4.2.4 Contribution of traffic sources

[revised manuscript text omitted]

26)    L638-642: Similarly, the attribution of 19% of BC to Euro 4 vehicles (15% of the fleet) is simplification that does not hold. The authors cire a paper (Jezek et al., 2018) which has shown that 2/3 of the BC emissions are caused by ¼ of the vehicles. Treating the emissions linearly is blatantly wrong. There are super-emitters in the fleet which contribute disproportionately and using the fleet composition as the argument to attribute emissions to Euro 4 vehicles is wrong. Additionally, this is an important conclusion (as noted by the authors – it is included in the abstract) and therefore requires an extended explanation.

As mentioned above, we have added a new section 4.2.4 Contribution of traffic sources to the main text. This section provides additional details on the methodology used to estimate exhaust emissions (lines 942-968).

The representative emission factors for the two simulated vehicle classes (light duty and heavy duty vehicles) were calculated by averaging the EFs of all subcategories, weighted by a combination of fleet composition and estimated annual mileage per vehicle class. This integrated approach allows the fleet composition to be adjusted to reflect the actual average presence of vehicles on the road. The same method is used to calculate the contribution of each sub-category (vehicle type, fuel and Euro standard) to the total emissions of both light and heavy duty vehicles. In addition, emission factors are calculated on the basis of vehicle speed to reflect the real traffic situation.

Our approach also takes into account super-emitters, such as pre-Euro1 diesel vehicles. However, their limited presence in the fleet and the low number of kilometres they travel per year result in a smaller contribution compared to other categories, such as Euro 4 diesel cars. Although the EFs of Euro 4 diesel passenger cars are lower than those of vehicles with older emission standards (Euro 3, Euro 2, etc.), their significant share in the actual fleet composition (number of registered vehicles adjusted by annual mileage per vehicle class) leads to a significant contribution from this vehicle class.

We changed the text as follows:
Lines 942-968
In order to further explain the percentage breakdown of the subcategories to the exhaust emissions, the methodology used to calculate different contributions is resumed. The representative EF for the two simulated vehicle classes (light and commercial vehicles) is calculated by averaging the EFs of all subcategories, weighted by a combination of the fleet composition and estimated annual mileage per vehicle class. This integrated approach allowed the composition of the registered fleet to be adjusted to reflect an actual average presence of vehicles on the road (see Figure S14 for the fleet composition resulting from this calculation). The same approach is then used to calculate the contribution of each subcategory (vehicle type, fuel and Euro standard) to the total emissions of either light or heavy duty vehicles. From this analysis, passenger cars (PC) are the main contributor to the exhaust component accounting for 35% of total transport emissions, while mopeds and motorcycles, light commercial vehicles, buses and trucks contribute 3%, 10% and 2% respectively. In addition, Euro 4 diesel passenger cars, following the aforementioned methodology, are responsible for the largest share of

BC emissions in the urban environment, accounting for 19% of total transport emissions.  Although their EFs are lower than those of vehicles with older emission standards (Euro 3, Euro 2, etc., see also Figure S15), their share in the actual fleet composition (number of registered vehicles adjusted by annual mileage per vehicle class) makes this vehicle class to account for a large contribution.

The uncertainty of the total exhaust contribution is estimated to be 16% based on the standard deviation of the three simulated scenarios considering different BC speciation factors (see Section 2.5.1). On the other hand, ~~BC emissions from tyre wear dominate the non-exhaust contribution, accounting for 20% of total traffic emissions, while brake wear and road wear emissions contribute 2% and 3% respectively. While these percentages represent emissions, it is important to note that all traffic emissions were consistently distributed over the same road areas and BC was treated as inert in the simulation. Consequently, the simulated BC concentrations closely match the percentages observed in the emission data. A summary of the percentage distribution of traffic emissions for the urban contribution can be found in the Sankey diagram shown in Figure 10.~~ Kouridis et al. (2010), suggested that the uncertainty of the exhaust EFs at a speed of 50 km h$^{-1}$ estimated following the Tier 3 approach, is around 0.001 g km$^{-1}$ for both petrol and diesel passenger cars (corresponding to 53% and 7% of the weighted EFs used for the same categories in this study), around 0.06 g km$^{-1}$ (8%) for commercial vehicles and about 0.076 g km$^{-1}$ (48%) for buses. No data is available for mopeds and motorcycles.

27) Supplement, Fig. S10: What is the source of the diurnal profiles? Were they measured? How?

The diurnal profiles for industrial activities (SNAP 3) and other mobile sources (SNAP 8) were sourced from the emission model Emisurf2020 (https://www.lmd.polytechnique.fr/chimere/), which is commonly used with the chemical transport model CHIMERE (Menut et al., 2021). The diurnal profile for non-industrial combustion (domestic heating activities) was derived from previous modeling studies focusing on the Po Valley (Veratti et al., 2023). The traffic diurnal profile for highways (SNAP 7 - highway) was deduced from direct vehicle counts conducted by Anas S.p.A., the Italian company responsible for road infrastructure, managing the network of national highways and motorways (https://www.stradeanas.it/it/le-strade/osservatorio-del-traffico/archivio-osservatorio-del-traffico). The urban traffic modulation profile (SNAP 7 - urban) was deduced from direct traffic measurements conducted in the city using data from historical traffic measurement data from 400 induction loops devices and radar Doppler data collected in winter 2016 (Ghermandi et al., 2019).

28) Supplement, Fig. S10: What is the source of the fleet composition? Please cite a reference.

The fleet composition, including the number of registered vehicles for each class, was obtained from the ACI website (http://www.aci.it/laci/studi-e-ricerche/dati-e-statistiche/autoritratto.html). The estimates for the number of kilometers traveled by each vehicle category were sourced from the ISPRA website (https://emissioni.sina.isprambiente.it/inventario-nazionale/). Both references have been added to the text (lines 387 and 388).

The availability of high resolution emission inventories produced by environmental agencies can sometimes be challenging. Therefore, a comparison between the annual BC emissions used in a study for the Emilia-Romagna region (INEMAR 2017) and common European and global emission datasets can provide valuable insights for future modelling studies in the Po valley. Table 4 presents a comparison between INEMAR 2017 (INEMAR, 2023; Marongiu et al., 2024), CAMS-REGv4.2 (2018; Kuenen et al., 2022), EDGARv6.1 (2018; Crippa et al., 2020) and EMEP (2017: Ullrich et al., 2023), categorised by sector. The comparison results show that for non-industrial combustion, EDGARv6.1 and EMEP agree quite well with INEMAR, with discrepancies between 8% and 10%, while CAMS-REGv4.2 is about 60% higher than INEMAR. For traffic emissions, the annual totals are very close to each other, with differences ranging between 3% and 22%, confirming that this sector is relatively well constrained. However, total emissions for other sectors exhibit significant discrepancies, partly due to the different classification systems used by different inventories to categorise emissions. Despite these differences, total BC emissions are relatively well aligned, with differences ranging from 11% to 17%.

**Table 4**. Comparison of annual BC emissions for the Emilia-Romagna region across different emission sectors as reported by INEMAR 2017, CAMS-REGv4.2, EDGARv6.1 and EMEP 2017.

| Area | Emission sector | INEMAR 2017 (tons) | CAMS-REGv4.2 (tons) | EDGARv6.1 (tons) | EMEP 2017 (tons) |
|---|---|---|---|---|---|
| Emilia-Romagna region | Non-industrial combustion | 672 | 1090 | 622 | 744 |
| | Industrial combustion | 30 | 65 | 154 | 68 |
| | Traffic | 532 | 549 | 586 | 680 |
| | Other mobile machinery | 298 | 109 | 1 | 300 |
| | Other sectors | 3 | 42 | 13 | 61 |
| | Total | 1535 | 1855 | 1376 | 1853 |

- Since the Eulerian approach is built on the CHIMERE model, I would suggest spending some more work on the previous evaluation studies performed on CHIMERE. The model has actively participated in numerous model intercomparisons exercises e.g., EURODELTAIII (Bessagnet et al., 2016) and AQMEII, and I think it would be beneficial to this study, and to the past modelling intercomparison efforts, to briefly comments on previous modelling results.

We appreciate the reviewer's suggestion and have included a detailed discussion of previous evaluation studies of the CHIMERE model in the manuscript. Specifically, we have reported the results of the main statistical metrics achieved by CHIMERE compared to observations during the simulation of EC, gaseous species (NO2, O3 and SO2), PM10 and PM2.5. This includes results from the EURODELTAIII (Bessagnet et al., 2016; Mircea et al., 2019) and POMI (Pernigotti et al., 2013) intercomparison exercises. The additional text has been inserted in lines 297-320:

The performance of CHIMERE in reproducing gaseous species, total PM and PM components has been extensively evaluated in several intercomparison studies. Notably, the EURODELTAIII (Bessagnet et al., 2016; Mircea et al., 2019), AQMEII (Pirovano et al., 2012), and POMI (Pernigotti et al., 2013) exercises have been significant in this context. Specifically, Mircea et al. (2019) focused on the simulation of carbonaceous aerosols over Europe. When EC concentrations within the $PM_{2.5}$ matrix were compared with corresponding measurements, CHIMERE showed satisfactory performance in reproducing the observed trend for the years 2006-2009, with a Pearson's correlation coefficient (r) between modelled and measured concentrations ranging from 0.6 to 0.85, a normalised centred Root Mean Square Error (RMSE) between 1.5 and 2.0 and a normalised Standard Deviation (SD) between 0.05 and 1.25. In the EURODELTAIII exercise (2006-2009), CHIMERE generally overestimated $O_3$ concentrations compared to other participating models, with a Mean Bias (MB)

between 6.3 and 22.5 µg m$^{-3}$. However, in 2009, all models, including CHIMERE, underestimated the measured concentrations due to biased boundary conditions. Despite this, CHIMERE had the lowest correlation coefficient (ranging from 0.27 to 0.71) but also the lowest RMSE among the models. For NO$_2$, its performance was comparable to other models, with r ranging from 0.67 to 0.72 and MB ranging from -0.64 to 0.64 µg m$^{-3}$, depending on the simulation year. For SO$_2$, the correlation coefficient ranged from 0.2 to 0.4, similar to other models, but CHIMERE was closest to the observations (MB from -0.46 to 0.13 µg m$^{-3}$). For PM$_{10}$, CHIMERE generally underestimated the measurements, as did other models, with MB ranging from -6.59 to -0.52 µg m$^{-3}$, but it achieved the highest correlation coefficient (0.7). Similarly, CHIMERE performed well for PM$_{2.5}$ over all simulated years, with the lowest RMSE (6.59 µg m$^{-3}$) and a comparable MB (between -1.00 and -2.39 µg m$^{-3}$) to the other models.

In addition, in the POMI intercomparison exercise, CHIMERE was used alongside EMEP, AURORA, CAMx, TCAM, and REM-CALGRID to simulate O$_3$ and PM over the Po Valley. The results indicated that CHIMERE's performance was comparable to the other models. For daily PM$_{10}$, CHIMERE had one of the lowest RMSE values (29.1 µg m$^{-3}$) and the highest correlation coefficient (0.6), with other statistical metrics in good agreement with the other models. However, for O$_3$, CHIMERE had the highest absolute MB (-9.1 µg m$^{-3}$) but also one of the lowest RMSE values (31.5 µg m$^{-3}$), with other metrics in agreement with the other models.

- I think there is no mention in the manuscript regarding how the BC mass is distributed over the size distribution. As far as I am aware, CHIMERE centres the distribution of BC at 200nm with a 1.2 sigma, which is a proper guess for background BC concentrations. Is it the case (and representative) also for this study? Or does the size distribution of BC differ between what is applied in the city and what is considered as background?

Yes, the reviewer is correct that the main text lacks this specific detail. As suggested, simulations with the CHIMERE model used a BC mass distribution centered at 200 nm, with a sigma of 1.2. This setup aligns with previously reported experimental studies analyzing BC distribution at various urban and rural sites (Li et al.. 2023; Ning et al. 2013; Schwarz et al., 2008). In contrast, the Lagrangian particle dispersion model GRAL uses a less sophisticated approach for simulating aerosols. Specifically, BC emissions are treated as PM$_{2.5}$, with deposition processes based on this particle size following the VDI guidelines (VDI 3945-3). Relevant details have been added to the main text at lines 279-280 and 364.

Lines 279-280:
Aerosols were represented using a size bin approach with 10 bins with mean mass median diameters ranging from 10 nm to 40 µm, and BC was simulated with a mass distribution centred at 200 nm with a sigma of 1.2, consistent with previously reported experimental studies (Schwarz et al., 2008; Ning et al., 2013; Li et al., 2023).

Line 364:
Lagrangian dispersion calculations were then performed at hourly intervals in prognostic mode, using the particle diameter of 2.5 µm.